# PiCSAR: Probabilistic Confidence Selection And Ranking for Reasoning Chains

## Abstract

Best-of-$n$ sampling improves the accuracy of large language models (LLMs) and large reasoning models (LRMs) by generating multiple candidate solutions and selecting the one with the highest reward. The key challenge for reasoning tasks is designing a scoring function that can identify correct reasoning chains without access to ground-truth answers. We propose **P**roba**b**ilistic **C**onfidence **S**election **A**nd **R**anking (PiCSAR): a simple, training-free method that scores each candidate generation using the joint log-likelihood of the reasoning and final answer. This method utilises both the scores of the reasoning path (*reasoning confidence*) and the final answer (*answer confidence*). PiCSAR achieves substantial gains across diverse benchmarks (+11.7 on AIME2024, +9.81 on AIME2025), outperforming baselines with fewer than at least 2x samples in 20 out of 25 comparisons. Our analysis reveals that correct reasoning chains exhibit significantly higher reasoning and answer confidence, justifying the effectiveness of PiCSAR.

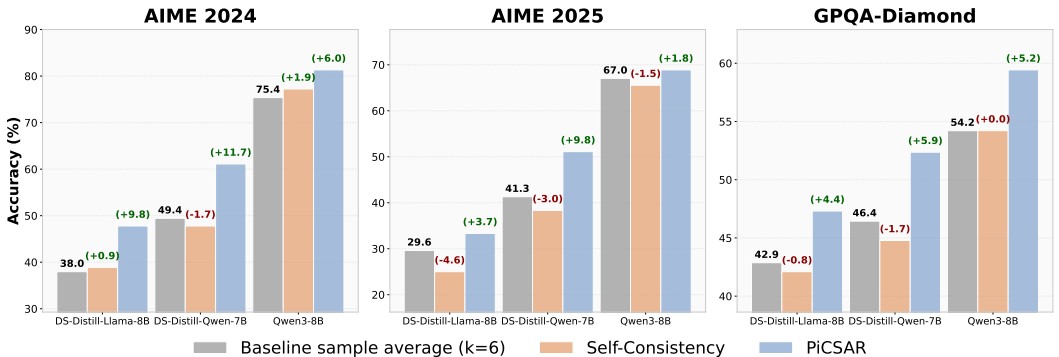

Figure 1: Performance of PiCSAR on three datasets (AIME 2024, AIME 2025, and GPQA-Diamond) and three models (DeeepSeek-Distill-Llama-8B, DeeepSeek-Distill-Qwen-7B, and Qwen3-8B), compared to self-consistency.

## 1 Introduction

Recent studies have shown that large language models (LLMs) achieve strong performance on complex reasoning tasks (Grattafiori et al., 2024; Team et al., 2024; Hurst et al., 2024); Techniques such as Chain of Thought (CoT, Wei et al., 2022; Kojima et al., 2022) aim to enhance the reasoning process, which generate explicit intermediate reasoning steps. Building on these advances, large reasoning models (LRMs) – LLMs that received intensive reasoning-focused post-training, such as OpenAI's o1 (Jaech et al., 2024), DeepSeek R1 (Guo et al., 2025), and Qwen3 (Yang et al., 2025a) – can solve relatively complex problems through long chains of thought, or a thinking process, often characterised as extended CoT with self-reflection (Yang et al., 2025b; Muennighoff et al., 2025).

Despite these advances, classic decoding approaches such as greedy decoding often fall short of state-of-the-art performance on complex benchmarks (Team et al., 2025; Balunović et al., 2025), emphasising the need for more sophisticated inference-time strategies. *Best-of-N* (BoN) sampling (Stiennon et al., 2020) emerged as an important technique, where $n$ candidate responses are generated

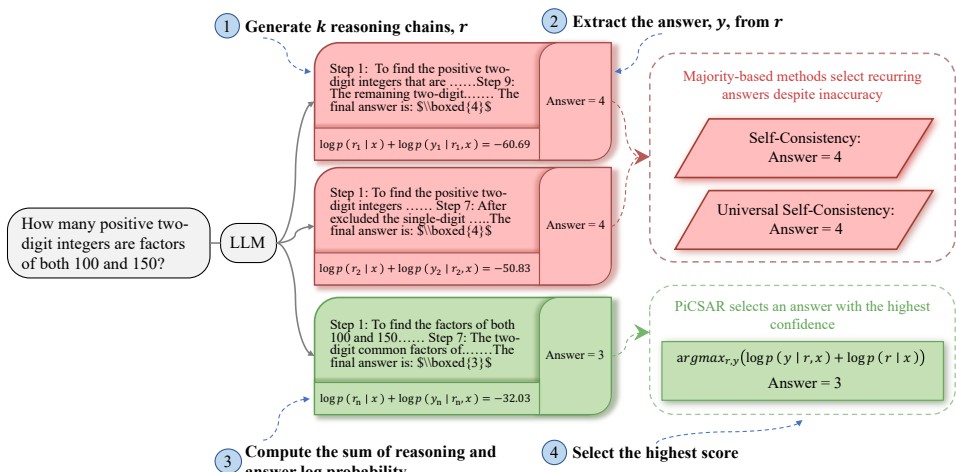

Figure 2: Example with *llama-3.1-8B* on *MATH500*, where PiCSAR selects the most likely reasoning trace $r$ and answer $y$ by jointly maximising their log-likelihoods $\log p(r \mid x)$ and $\log p(y \mid r, x)$.

and the one with the highest score from a reward model is selected (Mudgal et al., 2024; Huang et al., 2025). However, training or fine-tuning external reward models can be computationally expensive (Wang et al., 2023a) and can be vulnerable to distribution shifts (Eisenstein et al., 2023).

This led to the adoption of simpler, training-free BoN variants like Self-Consistency (Wang et al., 2023b), which selects the most frequent answer among multiple generated outputs. However, a key limitation of Self-Consistency is its exclusive reliance on the final answer while ignoring the reasoning that leads to it. Extensions like Universal Self-Consistency (USC, Chen et al., 2023b) prompt the model itself to identify the most consistent response from a set of candidates. USC, while evaluating complete responses, identifies the majority consensus pattern rather than the correctness of the reasoning; it discards valuable signals from the reasoning process itself, such as its coherence and plausibility, that contribute to reaching the answer. USC faces additional constraints from model context-window capacity and the reasoning ability of the model (Chen et al., 2023b), with Kang et al. (2025) showing that it is especially ineffective with smaller models. Attempts to overcome this by prompting the model to self-evaluate its reasoning verbally are often ineffective, as this form of explicit confidence can be poorly calibrated (Miao et al., 2024; Taubenfeld et al., 2025).

To address these challenges, we introduce Probabilistic Confidence Selection And Ranking (PiCSAR), a probabilistic confidence method for selecting a reasoning chain $r$ together with its answer $y$ without requiring any additional training or fine-tuning. Our approach is straightforward to implement and can be used with any LLM or LRM as an inference-time tool. It is based on a new scoring function that, given a prompt $x$, selects a reasoning chain $r$ and the answer $y$ via maximising their joint conditional likelihood $\log p(y, r \mid x)$. This objective naturally separates into two complementary components. The *reasoning confidence* term $\log p(r \mid x)$ promotes high-probability reasoning sequences by implicitly evaluating the likelihood of the chain given the prompt. The *answer confidence* term $\log p(y \mid r, x)$ quantifies the model's certainty in its final prediction, conditioned on the generated reasoning chain. Figure 2 shows a high-level outline of PiCSAR, and how it can solve instances that Self-Consistency and USC cannot solve correctly.

We evaluate PiCSAR on reasoning tasks across five LLMs and three LRMs, outperforming Self-Consistency and USC in most cases. PiCSAR achieves these improvements with substantially fewer samples, often requiring only $k = 6$ samples to outperform them even when using $k = 16$ or $32$ samples. In particular, PiCSAR manages to substancially improve the performance of LRMs, with Deepseek-R1-distilled-Llama-3 achieving *+13.33% and +12.78%* over Self-Consistency on AIME2024 and AIME2025, respectively (Figure 1). Unlike USC, which is bounded by the underlying model's reasoning abilities, PiCSAR allows confidence scores to be estimated by separate models. Even smaller models can approximate confidence effectively, as the evaluator captures stable properties of the reasoning process rather than artefacts themselves (Section 5.3).

---

**Algorithm 1** Probabilistic Confidence Selection And Ranking (PiCSAR)

---

1: **Input:** Prompt $x$, number of samples $k$, instruction prompt $\langle a \rangle$.
2: **Output:** Reasoning chain $r^*$ and answer $y^*$.
3: **Generate Candidates:** Independently sample $k$ reasoning chains $\{r_1, r_2, \ldots, r_k\}$ from the model, where each $r_i \sim p(r \mid x)$ for $i = i \ldots k$.
4: **Score Candidates:**
5: **for** each $i \in \{1, \ldots, k\}$ **do**
6:     **Extract Reasoning Confidence:** Retrieve $C_{\text{reason}}(i) = \log p(r_i \mid x)$ from generation $r_i$.
7:     **Extract Answer:** Extract answer, $y_i$, from reasoning chain, $r_i$.
8:     **Compute Answer Confidence:** Compute $C_{\text{answer}}(i) = \log p(y_i \mid \langle a \rangle, r_i, x)$.
9:     **Compute Final Score:** $\text{Score}(i) = C_{\text{reason}}(i) + C_{\text{answer}}(i)$.
10: **end for**
11: **Select Best:** Find the index of the highest-scoring candidate: $i^* = \arg\max_i \text{Score}(i)$.
12: **Return:** $(r_{i^*}, y_{i^*})$.

---

Beyond empirical results, we provide a comprehensive analysis of LLM confidence behaviour. At a finer granularity, we analyse answer confidence at a sentence level, using a peak-to-sentence ratio, which we term *information density*, that counts how often a reasoning chain attains high confidence relative to its length. We find that higher accuracy correlates with a high ratio, within the model family (Section 5.1). We show that answer confidence positively correlates with downstream accuracy. In addition, we demonstrate that confidence values are model-dependent and should not be used for direct comparison across models for ranking (Section 5.2).

## 2 A JOINT PROBABILISTIC METHOD FOR REASONING CHAIN SELECTION

We propose a training-free method for selecting a reasoning chain from a set of candidates, grounded in a probabilistic framework that leverages the model's confidence as its scoring signal. We frame the selection problem as an approximation of maximum a posteriori (MAP) decoding over the joint space of reasoning chains and final answers.

### 2.1 SCORING FUNCTION AND LOG-LIKELIHOOD DECOMPOSITION

We denote by $\mathcal{X}$ a set of possible prompts, $\mathcal{R}$ a set of reasoning chains, and $\mathcal{Y}$ the set of possible final answers. For a given input prompt $x \in \mathcal{X}$, our goal is to find the high confidence reasoning chain $r \in \mathcal{R}$ and its corresponding answer $y \in \mathcal{Y}$. Consider a selection criterion that aims to identify the pair $(r, y)$ with the highest joint conditional probability, $p(r, y \mid x)$. By the chain rule of probability, this decomposes into two distinct components:

$$p(r, y \mid x) = p(y \mid r, x) \cdot p(r \mid x). \tag{1}$$

In log-space, the joint probability becomes the sum of two log-likelihood terms as follows:

$$\text{Score}(r, y) = \underbrace{\log p(r \mid x)}_{\text{Reasoning Confidence}} + \underbrace{\log p(y \mid r, x)}_{\text{Answer Confidence}}. \tag{2}$$

These two terms provide complementary signals regarding the quality of a candidate generation:

- **Reasoning Confidence** ($\log p(r \mid x)$)**:** This term quantifies the model's confidence in generating $r$ given the prompt $x$. It quantifies the plausibility of the reasoning path itself.
- **Answer Confidence** ($\log p(y \mid r, x)$)**:** This term measures the model's certainty in the final answer $y$, *conditioned on the specific reasoning chain it has produced*.

### 2.2 PROBABILISTIC CONFIDENCE SELECTION AND RANKING (PiCSAR)

Directly selecting $r \in \mathcal{R}$, $y \in Y$, where the joint log likelihood $\text{Score}(r, y)$ is maximised over the unconstrained space of possible pairs, is intractable. We therefore approximate this optimisation with our PiCSAR sampling-based approach, as outlined in Algorithm 1. We first generate a set of $k$ candidate reasoning chains $r_i \in \{r_1, r_2, \ldots r_k\}$ from the model's posterior $p(r \mid x)$. Each chain

$r_i$ implies a corresponding final answer $y_i$. We then re-rank these candidates using our PiCSAR scoring function.

The *reasoning confidence* term is obtained by summing the token-level log-probabilities from the model during the generation of $r_i$. By not applying length normalisation, this term naturally favours more concise and direct reasoning paths as it involves a cumulative sum of individual token log-probabilities. We also consider the length-normalised variant, PiCSAR-N, which focuses more on the impact of log probability per token rather than favouring concise reasoning paths, leading to similar results. (Details and results in Appendix C.3.)

The *answer confidence* term, $\log p(y \mid r, x)$, however, presents a practical challenge. As the model's distribution is over all possible text continuations, the probability of a final answer is confounded by the likelihood of whatever text might follow it. This makes the raw log-probabilities of different answers fundamentally incomparable. To address this and ensure we can reliably extract a final answer for answer confidence computation, we condition the model on an explicit instruction prompt, denoted as $\langle a \rangle$, which is appended after the reasoning chain. This prompt explicitly asks the model to provide the final answer based on the preceding context (*i.e.,* "*When you see a potential reasoning followed by $\langle$sep$\rangle$, output the final answer.*"), with details of the prompt provided in Appendix B. While we extract the answer $y$ directly from the reasoning chain $r$, we use this augmented prompt to compute the answer confidence. Our modified objective is thus:

$$\arg \max_{r,y} \left[ \log p(r \mid x) + \log p(y \mid \langle a \rangle, r, x) \right]. \tag{3}$$

This modification grounds the answer confidence computation squarely in the reasoning provided, allowing for a more targeted estimation of answer confidence.

The final step is to select the candidate pair with the highest score. As illustrated in Figure 2, the two components of our scoring function play complementary roles. The *reasoning confidence* is the sum of log-probabilities for every token in the reasoning chain. Since these log-probabilities are negative, their sum naturally accumulates to a larger negative magnitude for longer sequences (as shown in Figure 3). It thereby acts as a coarse-grained filter, placing strong selective pressure on the overall plausibility of the reasoning process itself. The *answer confidence* then serves as a powerful, fine-grained discriminator, often proving decisive when multiple candidate chains exhibit similar reasoning plausibility. Consequently, the summation of these two components constitutes the **joint log-likelihood** of the entire trajectory: $\log p(r, y|x) = \log p(r|x) + \log p(y|r, x)$. Rather than discarding sequence-level information, this formulation explicitly aggregates the autoregressive dependencies into a unified metric. By maximising this joint probability via re-ranking, PiCSAR can identify the trajectory that simultaneously ensures reasoning plausibility and answer certainty, thereby filtering for the most likely correct reasoning chain.

### 2.3 MOTIVATION: CONFIDENCE INFORMATION PLANE

To motivate PiCSAR design, we analyse the distribution of model-generated samples on a 2D "Information Plane", with respect to our two confidence terms (Figure 3). We partition the plane into four quadrants using the median value of each axis. $\log p(y \mid r, X) = -10$ represents the value used as a "fallback mechanism" when the model fails to answer, (*i.e.,* when no answer token is generated and the answer-confidence term cannot be computed.) We tested this fallback mechanism with various values, and the results are in Appendix C.7. For Llama-3.1-8B on the MATH500 dataset, a striking pattern emerges: correct answers (green) are concentrated in the upper-right quadrant (Q1), corresponding to high scores on *both* confidence terms.

The quadrant-wise accuracy breakdown is stark: the upper-right quadrant (Q1) achieves 71.7% accuracy, outperforming other quadrants (Q2: 39.0%, Q3: 31.6%, Q4: 62.2%). High reasoning confidence (Q1 and Q4) leads to

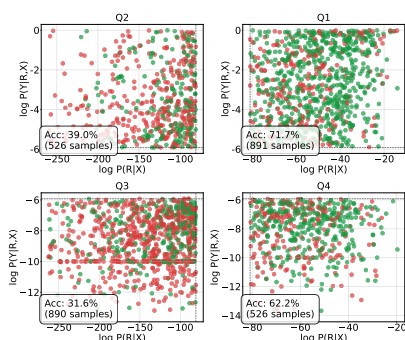

Figure 3: Information plane of MATH500 questions that Llama-3.1-8B predicts correctly and incorrectly ($k = 6$). Quadrants show combinations of answer and reasoning confidence. This pattern is consistent across LLMs, LRMs, and datasets (Appendix E).

a higher performance than a high answer confidence (Q2 and Q3). This is reinforced by a statistical t-test that, while both terms are highly significant predictors of correctness, reasoning confidence is a significantly stronger predictor (t-statistics $\approx 9.111$) than answer confidence (t-statistics $\approx 4.753$). For more details on the statistical tests, see Appendix E.2. Nevertheless, both answer and reasoning confidence measures remain essential components for reasoning chain selection.

This principle can be used as a practical filter; tightening the thresholds to the 75th percentile, for instance, isolates a subset of samples with near-perfect accuracy (*i.e.,* 100% on DS-Distilled-Qwen-2.5-7B with AIME2025), providing a mechanism to identify reliably instances (More examples and datasets can be referred to Appendix E). *Overall, our analysis reveals that correct reasoning tends to have higher reasoning and answer confidence, with reasoning confidence being a substantially stronger predictor of correctness.*

## 3 EXPERIMENTAL SETUP

**Models**   To demonstrate the generalisability of our approach, we conduct evaluations across a diverse set of recent LLMs and LRMs. Our experiments include LLMs from three major families: Llama-3.1-Instruct (8B and 70B; Dubey et al. 2024), Gemma-2-Instruct (9B; Team et al. 2024), and Qwen3 (8B and 32B; Yang et al. 2025a). For the Qwen3 models, we disable the *thinking mode* for fair comparison. For LRMs, we include two distilled models from the DeepSeek-R1 series (DS-distill-Llama-3.1-8B and DS-distill-Qwen-2.5b; Guo et al. 2025), and the Qwen-3-8B model with *thinking mode* enabled. We exclude larger LRMs due to computational cost.

**Baseline Methods**   We compare PiCSAR against six baselines: (1) *Greedy Decoding*; (2) *Self-Consistency* (Wang et al., 2023b); (3) *Universal Self-Consistency* (Chen et al., 2023b)); (4) *p(True)* (Kadavath et al., 2022); (5) *Self-Certainty* (Kang et al., 2025). We include (6) *Confidence-Informed Self-Consistency (CISC)* Taubenfeld et al. (2025) in Appendix C.1, as it mainly involves weight voting. CISC originally proposed with weight voting through *p(True)*, while we include a comparison with CICS (PiCSAR) for fair comparison. Due to context length limitations and computational constraints, we exclude (3), (4) and (5) in LRMs and $k = 16, 32$ in LLMs.

To isolate the contribution of each component in PiCSAR, we include three ablations in Appendix C.2 and C.3: *Reasoning Confidence* ($\max_r(\log p(r \mid x))$), with (6), and without (7) length normalisation respectively, and (8) *Answer Confidence* ($\max_y(\log p(y \mid r, x))$). For LRMs, we compare against (1), (2), (6), (7), and (8). For all datasets, we include the $pass@k$ upper bound, representing the maximum achievable accuracy when at least one of the $k$ candidates is correct. Implementation details can be found in Appendix B.

**Datasets and Evaluation Metrics**   We evaluate on five benchmarks for LLMs, with three mathematical benchmarks: GSM8K (Cobbe et al., 2021), SVAMP (Patel et al., 2021), and MATH500 (Hendrycks et al., 2021), and two general scientific reasoning benchmarks: GPQA-Diamond (Rein et al., 2024) and TheoremQA Chen et al. (2023a). For LRMs, we additionally evaluate on AIME2024 and 2025, which are omitted from the LLM setting given their difficulty. All results averaged over three independent runs and reported with standard errors.

## 4 EXPERIMENTAL RESULTS

### PERFORMANCE ON LARGE LANGUAGE MODELS

Based on Table 1, we analyse our results based on the LLM model families. Llama models (Llama-3.1-8B and 70B) show consistent improvements across all baselines. With $k = 6$ sampling, Llama-3.1-8B outperforms the best-performing baseline (*i.e.,* Self-Certainty) by 3.26% of average accuracy score (26.54% $\rightarrow$ 29.80%) on GPQA-Diamond. Llama-3.1-70B demonstrates similar gains: 7.07% improvement over Self-Certainty and 5.66% over USC. We can also observe a similar trend on Gemma-2-9B. At $k$=6, PiCSAR outperforms Self-Consistency by 4.93%. This outcome aligns with our information-plane analysis (see Figure 3); PiCSAR selects candidates in the top-right, high-accuracy quadrant by maximising the joint score of reasoning and answer confidence.

| Method | SVAMP | | GSM8K | | MATH500 | | GPQA-Diamond | | TheoremQA | |
|---|---|---|---|---|---|---|---|---|---|---|
| | $k=6$ | $k=16/32$ | $k=6$ | $k=16/32$ | $k=6$ | $k=16/32$ | $k=6$ | $k=16/32$ | $k=6$ | $k=16/32$ |
| *Gemma-2-9B-Instruct* | | | | | | | | | | |
| Greedy Decoding | 87.33 | | 86.64 | | 41.40 | | 29.80 | | 17.14 | |
| Self-Consistency | $88.15_{\pm0.22}$ | $88.89_{\pm0.22}$ | $87.04_{\pm0.24}$ | $88.10_{\pm0.05}$ | $41.60_{\pm0.40}$ | $43.27_{\pm0.23}$ | $27.27_{\pm0.58}$ | $23.91_{\pm1.38}$ | $15.44_{\pm0.12}$ | $14.10_{\pm0.00}$ |
| USC | $88.63_{\pm0.13}$ | - | $85.74_{\pm0.27}$ | - | $42.54_{\pm0.37}$ | - | $24.33_{\pm1.21}$ | - | $17.24_{\pm0.33}$ | - |
| p(True) | $88.56_{\pm0.44}$ | $87.89_{\pm0.22}$ | $88.36_{\pm0.22}$ | $88.38_{\pm0.08}$ | $\mathbf{46.87}_{\pm0.07}$ | $46.80_{\pm0.70}$ | $30.30_{\pm1.54}$ | $33.50_{\pm0.17}$ | $15.62_{\pm0.37}$ | $15.98_{\pm0.44}$ |
| Self-Certainty | $88.48_{\pm0.04}$ | $88.33_{\pm0.06}$ | $87.18_{\pm0.08}$ | $87.32_{\pm0.03}$ | $43.93_{\pm0.13}$ | $43.93_{\pm0.08}$ | $26.77_{\pm0.42}$ | $27.41_{\pm0.83}$ | $14.73_{\pm0.28}$ | $14.77_{\pm0.04}$ |
| PiCSAR | $\mathbf{89.00}_{\pm0.38}^{*}$ | $\mathbf{91.02}_{\pm0.59}$ | $\mathbf{88.66}_{\pm0.11}^{*}$ | $\mathbf{88.99}_{\pm0.20}$ | $46.53_{\pm0.29}^{*}$ | $\mathbf{47.13}_{\pm0.13}$ | $\mathbf{32.32}_{\pm0.51}^{*}$ | $\mathbf{34.01}_{\pm1.94}$ | $\mathbf{18.62}_{\pm0.39}^{*}$ | $\mathbf{18.88}_{\pm0.54}$ |
| Upper Bound | $93.44_{\pm0.22}$ | $95.67_{\pm0.38}$ | $93.44_{\pm0.09}$ | $95.60_{\pm0.04}$ | $58.47_{\pm0.27}$ | $66.67_{\pm0.47}$ | $55.22_{\pm1.10}$ | $82.49_{\pm1.02}$ | $24.32_{\pm0.49}$ | $32.40_{\pm0.20}$ |
| *Llama-3.1-8B-Instruct* | | | | | | | | | | |
| Greedy Decoding | 89.67 | | 87.47 | | 50.40 | | 27.27 | | 17.80 | |
| Self-Consistency | $88.33_{\pm0.67}$ | $89.89_{\pm0.11}$ | $86.67_{\pm0.38}$ | $89.52_{\pm0.16}$ | $46.33_{\pm0.13}$ | $50.13_{\pm0.48}$ | $26.09_{\pm0.45}$ | $26.67_{\pm1.34}$ | $15.62_{\pm0.18}$ | $12.72_{\pm0.48}$ |
| USC | $89.87_{\pm0.23}$ | - | $88.22_{\pm0.23}$ | - | $51.80_{\pm1.25}$ | - | $25.67_{\pm1.54}$ | - | $18.88_{\pm0.31}$ | - |
| p(True) | $85.33_{\pm0.00}$ | $83.22_{\pm0.91}$ | $87.40_{\pm0.44}$ | $86.59_{\pm0.03}$ | $47.73_{\pm0.66}$ | $47.80_{\pm0.72}$ | $27.27_{\pm1.75}$ | $26.09_{\pm2.07}$ | $14.41_{\pm0.59}$ | $14.10_{\pm0.51}$ |
| Self-Certainty | $89.44_{\pm0.06}$ | $89.49_{\pm0.26}$ | $87.43_{\pm0.24}$ | $87.35_{\pm0.02}$ | $51.04_{\pm0.20}$ | $51.09_{\pm0.16}$ | $26.54_{\pm0.49}$ | $26.30_{\pm0.49}$ | $14.91_{\pm0.13}$ | $14.62_{\pm0.14}$ |
| PiCSAR | $\mathbf{91.78}_{\pm0.11}^{*}$ | $\mathbf{93.44}_{\pm0.89}$ | $\mathbf{89.09}_{\pm0.13}^{*}$ | $\mathbf{89.98}_{\pm0.23}$ | $\mathbf{53.33}_{\pm0.73}^{*}$ | $\mathbf{53.87}_{\pm0.70}$ | $\mathbf{29.80}_{\pm1.34}^{*}$ | $\mathbf{33.67}_{\pm3.06}$ | $\mathbf{20.08}_{\pm0.43}^{*}$ | $\mathbf{19.72}_{\pm0.39}$ |
| Upper Bound | $96.78_{\pm0.11}$ | $99.11_{\pm0.11}$ | $96.15_{\pm0.07}$ | $98.18_{\pm0.04}$ | $72.80_{\pm0.23}$ | $82.20_{\pm0.60}$ | $65.82_{\pm1.50}$ | $92.76_{\pm0.73}$ | $28.20_{\pm0.32}$ | $37.84_{\pm1.13}$ |
| *Qwen3-8B (Non-thinking)* | | | | | | | | | | |
| Greedy Decoding | 93.33 | | 92.48 | | 73.40 | | 42.23 | | 27.71 | |
| Self-Consistency | $92.52_{\pm0.33}$ | $93.11_{\pm0.11}$ | $92.29_{\pm0.13}$ | $91.69_{\pm0.11}$ | $73.00_{\pm0.23}$ | $72.27_{\pm0.00}$ | $47.47_{\pm0.29}$ | $40.74_{\pm1.61}$ | $28.33_{\pm0.31}$ | $28.51_{\pm0.33}$ |
| USC | $93.11_{\pm0.22}$ | - | $\mathbf{93.24}_{\pm0.13}$ | - | $73.60_{\pm0.12}$ | - | $\mathbf{48.38}_{\pm2.06}$ | - | $27.88_{\pm0.55}$ | - |
| p(True) | $92.44_{\pm0.56}$ | $91.78_{\pm0.44}$ | $92.10_{\pm0.00}$ | $91.22_{\pm0.18}$ | $72.67_{\pm0.24}$ | $71.20_{\pm0.60}$ | $41.25_{\pm1.71}$ | $36.20_{\pm1.44}$ | $27.84_{\pm0.18}$ | $28.28_{\pm0.13}$ |
| Self-Certainty | $92.63_{\pm0.21}$ | $92.83_{\pm0.04}$ | $92.29_{\pm0.07}$ | $92.25_{\pm0.04}$ | $71.94_{\pm0.16}$ | $71.82_{\pm0.14}$ | $44.33_{\pm0.54}$ | $42.29_{\pm0.81}$ | $27.97_{\pm0.66}$ | $27.92_{\pm0.77}$ |
| PiCSAR | $\mathbf{93.56}_{\pm0.22}^{*}$ | $\mathbf{95.13}_{\pm0.22}$ | $92.33_{\pm0.13}^{*}$ | $\mathbf{93.22}_{\pm0.08}$ | $\mathbf{73.67}_{\pm0.24}^{*}$ | $\underline{73.40}_{\pm0.13}$ | $46.98_{\pm1.01}^{*}$ | $\mathbf{43.69}_{\pm1.26}$ | $\mathbf{29.76}_{\pm0.58}^{*}$ | $\mathbf{29.17}_{\pm0.64}$ |
| Upper Bound | $96.33_{\pm0.67}$ | $97.89_{\pm0.11}$ | $95.52_{\pm0.00}$ | $96.84_{\pm0.03}$ | $81.13_{\pm0.44}$ | $83.53_{\pm0.24}$ | $76.26_{\pm1.62}$ | $86.36_{\pm0.29}$ | $34.94_{\pm0.00}$ | $40.03_{\pm0.35}$ |
| *Llama-3.1-70B-Instruct* | | | | | | | | | | |
| Greedy Decoding | 94.33 | | 93.93 | | 60.20 | | 40.44 | | 30.79 | |
| Self-Consistency | $92.78_{\pm0.56}$ | $93.45_{\pm0.11}$ | $94.00_{\pm0.10}$ | $93.98_{\pm0.13}$ | $58.60_{\pm0.46}$ | $60.80_{\pm0.87}$ | $42.59_{\pm1.02}$ | $37.54_{\pm0.67}$ | $26.55_{\pm0.47}$ | $25.61_{\pm0.00}$ |
| USC | $92.78_{\pm0.11}$ | - | $93.29_{\pm0.20}$ | - | $60.60_{\pm0.95}$ | - | $41.25_{\pm1.76}$ | - | $27.44_{\pm0.67}$ | - |
| p(True) | $93.11_{\pm0.78}$ | $93.11_{\pm0.40}$ | $94.51_{\pm0.13}$ | $94.08_{\pm0.23}$ | $61.47_{\pm1.14}$ | $62.33_{\pm1.16}$ | $41.25_{\pm1.61}$ | $42.09_{\pm2.21}$ | $24.45_{\pm0.31}$ | $24.23_{\pm0.61}$ |
| Self-Certainty | $93.02_{\pm0.30}$ | $93.84_{\pm0.01}$ | $94.01_{\pm0.13}$ | $93.94_{\pm0.05}$ | $61.82_{\pm0.08}$ | $61.70_{\pm0.14}$ | $39.84_{\pm0.88}$ | $38.87_{\pm0.47}$ | $24.43_{\pm0.18}$ | $24.56_{\pm0.11}$ |
| PiCSAR | $94.10_{\pm0.11}^{*}$ | $\mathbf{95.58}_{\pm0.22}$ | $\mathbf{94.58}_{\pm0.03}^{*}$ | $\mathbf{94.81}_{\pm0.13}$ | $\mathbf{63.67}_{\pm1.51}^{*}$ | $\mathbf{64.07}_{\pm0.87}$ | $\mathbf{46.91}_{\pm2.65}^{*}$ | $\mathbf{46.46}_{\pm2.59}$ | $27.84_{\pm0.19}^{*}$ | $\mathbf{26.73}_{\pm0.27}$ |
| Upper Bound | $97.22_{\pm0.22}$ | $97.78_{\pm0.22}$ | $96.91_{\pm0.03}$ | $97.44_{\pm0.03}$ | $77.07_{\pm0.47}$ | $81.67_{\pm0.18}$ | $75.59_{\pm0.61}$ | $87.71_{\pm0.45}$ | $40.70_{\pm0.20}$ | $43.47_{\pm0.18}$ |
| *Qwen3-32B (Non-thinking)* | | | | | | | | | | |
| Greedy decoding | 92.33 | | 93.24 | | 75.00 | | 48.48 | | 29.99 | |
| Self-consistency | $92.67_{\pm0.33}$ | $93.11_{\pm0.33}$ | $93.62_{\pm0.00}$ | $93.75_{\pm0.08}$ | $75.93_{\pm0.33}$ | $\mathbf{76.27}_{\pm0.12}$ | $47.31_{\pm1.98}$ | $44.44_{\pm0.51}$ | $30.79_{\pm0.00}$ | $30.92_{\pm0.28}$ |
| USC | $92.44_{\pm0.78}$ | - | $93.69_{\pm0.13}$ | - | $76.16_{\pm0.64}$ | - | $44.90_{\pm0.55}$ | - | $30.07_{\pm0.51}$ | - |
| p(True) | $\mathbf{93.22}_{\pm0.11}$ | $93.00_{\pm0.69}$ | $92.79_{\pm0.53}$ | $92.91_{\pm0.25}$ | $74.07_{\pm1.07}$ | $74.00_{\pm0.35}$ | $39.90_{\pm2.81}$ | $38.05_{\pm0.94}$ | $30.79_{\pm0.00}$ | $30.08_{\pm0.12}$ |
| Self-certainty | $92.63_{\pm0.18}$ | $92.92_{\pm0.16}$ | $92.29_{\pm0.03}$ | $93.45_{\pm0.02}$ | $71.94_{\pm0.09}$ | $75.68_{\pm0.10}$ | $43.07_{\pm1.16}$ | $43.39_{\pm0.73}$ | $30.23_{\pm0.00}$ | $30.61_{\pm0.13}$ |
| PiCSAR | $\mathbf{93.22}_{\pm0.22}^{*}$ | $\mathbf{93.55}_{\pm0.33}$ | $\mathbf{93.90}_{\pm0.28}^{*}$ | $\mathbf{93.88}_{\pm0.22}$ | $\mathbf{77.00}_{\pm0.18}^{*}$ | $75.93_{\pm0.13}$ | $46.91_{\pm1.02}^{*}$ | $44.44_{\pm2.28}$ | $\mathbf{31.46}_{\pm0.04}^{*}$ | $\mathbf{31.42}_{\pm0.27}$ |
| Upper Bound | $96.78_{\pm0.11}$ | $98.00_{\pm0.00}$ | $96.28_{\pm0.13}$ | $96.99_{\pm0.07}$ | $82.27_{\pm0.13}$ | $83.73_{\pm0.07}$ | $72.56_{\pm1.87}$ | $86.20_{\pm1.02}$ | $39.76_{\pm0.00}$ | $42.93_{\pm0.12}$ |

Table 1: **Comparison of model accuracies across various baselines and benchmarks on LLMs.** Sampling: $k = \{6, 32\}$ for Gemma-2-9B, Llama-3.1-8B, Qwen3-8B; $k = \{6, 16\}$ for Llama-3.1-70B, Qwen3-32B due to computational constraints. **Bold**: highest, underline: equal highest, $^{*}$: $k = 6$ outperforms $k = 16, 32$ baselines. *PiCSAR with $k = 6$ outperforms larger $k$ in 20/25 cases.*

For the Qwen family (Qwen3-8B and Qwen3-32B), PiCSAR generally leads across benchmarks and sample counts ($k$). While there are a few exceptions, PiCSAR maintains the strongest overall profile. For instance, on MATH500 with $k = 6$, it improves the accuracy of Qwen3-8B from 75.93% (Self-

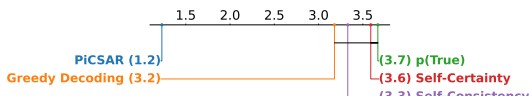

Figure 4: Critical Difference Diagram based on Nemenyi Test, $p < 0.05$.

Consistency) to 77.00%. Our results show that PiCSAR outperforms most existing baselines and datasets, demonstrating consistent improvements across various reasoning tasks. As shown in Appendix C.1, CISC (PiCSAR) consistently outperforms CISC (*p(True)*) across all baselines, indicating its potential for weighting augmentation, but detailed voting strategy analysis remains future work. As shown in Table 4, the Nemenyi post-hoc test Demšar (2006) confirms PiCSAR significantly outperforms all baselines ($p < 0.05$), achieving the best average rank of 1.2. *These findings validate our hypothesis that the model's confidence provides more informative signals than frequency-based selection.*

*PiCSAR is sample efficient.* PiCSAR with a small sampling budget ($k = 6$) frequently outperforms both Self-Consistency and Self-Certainty at higher sampling budgets ($k = 16, 32$), narrowing the gap to the upper bound by detecting correct reasoning even within a small sample. For instance, Gemma-2-9B Instruct with $k = 6$ (46.53%) outperforms $k = 32$ (43.27%). This indicates that correct reasoning chains are often present in small candidate sets, and that better selection is more important than increased sampling. (See Appendix C.6 for details of the upper bound analysis.)

Overall, the joint score acts as a paired scoring function: the *reasoning confidence* ($\log p(r \mid x)$), calculated over the full reasoning path, provides an assessment of plausibility towards its own reasoning, while the *answer confidence* ($\log p(y \mid r, x)$), focused on the final answer, serves as a fine-grained discriminator. This approach yields consistent improvements across evaluated models.

| Method | SVAMP | GSM8K | MATH500 | GPQA-Diamond | TheoremQA | AIME 2024 | AIME 2025 |
|---|---|---|---|---|---|---|---|
| *DS-Distill-llama-3-8B* | | | | | | | |
| Average | $82.11_{\pm0.13}$ | $73.67_{\pm0.32}$ | $65.55_{\pm0.25}$ | $42.87_{\pm1.07}$ | $26.58_{\pm0.06}$ | $37.96_{\pm1.52}$ | $29.63_{\pm0.37}$ |
| Self-Consistency | $\mathbf{86.17}_{\pm0.27}$ | $74.01_{\pm0.70}$ | $66.25_{\pm0.40}$ | $42.10_{\pm1.77}$ | $27.98_{\pm0.87}$ | $38.89_{\pm1.67}$ | $25.00_{\pm0.37}$ |
| PiCSAR | $85.67_{\pm0.07}$ | $\mathbf{76.42}_{\pm0.16}$ | $\mathbf{67.20}_{\pm0.60}$ | $\mathbf{47.31}_{\pm0.17}$ | $\mathbf{28.02}_{\pm0.78}$ | $\mathbf{47.78}_{\pm4.01}$ | $\mathbf{33.33}_{\pm1.11}$ |
| Upper Bound | $95.67_{\pm0.00}$ | $92.91_{\pm0.35}$ | $82.00_{\pm0.13}$ | $77.27_{\pm0.77}$ | $36.37_{\pm2.83}$ | $66.67_{\pm5.09}$ | $51.11_{\pm1.11}$ |
| *DS-Distill-Qwen-2.5-7B* | | | | | | | |
| Average | $89.26_{\pm0.13}$ | $87.29_{\pm0.14}$ | $72.79_{\pm0.16}$ | $46.44_{\pm1.63}$ | $33.11_{\pm0.14}$ | $49.44_{\pm3.06}$ | $41.30_{\pm1.30}$ |
| Self-Consistency | $90.39_{\pm0.20}$ | $\mathbf{89.50}_{\pm0.37}$ | $73.87_{\pm0.25}$ | $44.78_{\pm1.83}$ | $35.88_{\pm0.35}$ | $47.78_{\pm3.40}$ | $38.33_{\pm3.34}$ |
| PiCSAR | $\mathbf{91.78}_{\pm0.48}$ | $88.18_{\pm0.07}$ | $\mathbf{74.00}_{\pm0.70}$ | $\mathbf{52.36}_{\pm2.88}$ | $\mathbf{36.76}_{\pm0.44}$ | $\mathbf{61.11}_{\pm1.11}$ | $\mathbf{51.11}_{\pm1.11}$ |
| Upper Bound | $96.33_{\pm0.38}$ | $96.79_{\pm0.13}$ | $83.33_{\pm0.18}$ | $79.12_{\pm2.07}$ | $48.59_{\pm0.08}$ | $72.22_{\pm1.11}$ | $70.00_{\pm0.00}$ |
| *Qwen3-8B* | | | | | | | |
| Average | $91.43_{\pm0.07}$ | $95.43_{\pm0.01}$ | $80.44_{\pm0.10}$ | $54.21_{\pm0.83}$ | $40.83_{\pm0.13}$ | $75.37_{\pm0.19}$ | $67.04_{\pm2.06}$ |
| Self-Consistency | $91.83_{\pm0.33}$ | $95.68_{\pm0.03}$ | $80.40_{\pm0.18}$ | $54.21_{\pm1.68}$ | $41.81_{\pm0.11}$ | $77.23_{\pm1.11}$ | $65.56_{\pm2.58}$ |
| PiCSAR | $\mathbf{94.33}_{\pm0.33}$ | $\mathbf{95.94}_{\pm0.04}$ | $\mathbf{80.60}_{\pm0.13}$ | $\mathbf{59.43}_{\pm1.61}$ | $\mathbf{42.57}_{\pm0.27}$ | $\mathbf{81.33}_{\pm1.34}$ | $\mathbf{68.89}_{\pm2.22}$ |
| Upper Bound | $97.56_{\pm0.11}$ | $97.54_{\pm0.03}$ | $84.00_{\pm0.12}$ | $80.13_{\pm0.45}$ | $44.71_{\pm1.34}$ | $87.78_{\pm1.11}$ | $82.22_{\pm1.11}$ |

Table 2: **Comparison of model accuracies across various baselines and benchmarks on LRMs**. For all evaluations, we use $k = 6$ sampling. *PiCSAR outperforms 19/21 baselines and comparisons.*

## PERFORMANCE ON LARGE REASONING MODELS

Table 2 reports results on baselines evaluated from LRMs, with an additional of AIME 2024 and AIME 2025. We observe that PiCSAR outperforms all baselines across all 18 comparisons. Relative to Self-Consistency, DS-Distill-Llama-3-8B demonstrates substantial 8.89% improvements on AIME2024 and 8.33% on AIME2025. DS-Distill-Qwen-2.5-7B shows greater improvements compared to Self-Consistency, with 12.33% and 12.78% accuracy improvement on AIME2024 and AIME2025, respectively. When applied on a relatively more capable model such as Qwen3-8B, PiCSAR increases accuracy by 4.1% and 3.33% on AIME 2024 and AIME 2025, respectively. While improvements on previously evaluated benchmarks such as MATH500, SVAMP, and GSM8K yield smaller gains, we observe substantial improvements on GPQA-Diamond, with increases of 5.21%, 7.58%, and 5.22% for DS-Distill-Llama-3-8B, DS-Distill-Qwen-2.5-7B, and Qwen3-8B, respectively. These trends mirror those observed with LLMs: gains are most pronounced on challenging datasets where the models' initial baseline accuracies are relatively lower. *We conclude that PiCSAR, by jointly maximising reasoning and answer confidence, validates the information plane principle in Section 2.3 and provides a scoring method that improves accuracy both for LLMs and LRMs.*

## 5 ANALYSIS

In our analysis we focus on studying: (1) the peak-to-sentence ratio dynamics, analysing how the information density – the density of high-confidence steps in reasoning chains, correlates with overall accuracy; (2) the relationship between confidence scores and accuracy, both within and across models; (3) the robustness of our confidence metric when generation and evaluation are decoupled.

### 5.1 SENTENCE-LEVEL CONFIDENCE DYNAMICS AS A PROXY FOR REASONING QUALITY

To understand the dynamics of PiCSAR, we analyse the evolution of answer confidence across reasoning chains. For a given reasoning chain $r$ composed of sentences $(r^1, r^2, \ldots, r^m)$ and its corresponding final answer $y$, we measure how the model's confidence in $y$ changes as it processes more of the reasoning. We compute a sequence of scores, $\log p(y \mid r^{1:j}, x)$, for each partial reasoning prefix $r^{1:j}$, where $j$ ranges from 1 to $m$. To capture the characteristics of these confidence sequences, we rank the responses by PiCSAR scoring function into three groups (highest, middle, lowest), and analyse the "peakiness" of the confidence trajectory within each group. We define a *peak* as a sentence where the confidence $\log p(y \mid r^{1:j}, x)$ exceeds the 95th percentile of all sentence-level scores observed across reasoning chains with the correct answer for that particular problem. The *peak-to-sentence ratio* is the peak count divided by the total sentences. We term this *information density*: the proportion of reasoning sentences contributing meaningfully to answer confidence.

Table 3 reveals two key insights. (1) Higher peak-to-sentence ratio aligns with higher accuracy across different models, showing that *reasoning chains that lead to the correct answer tend to have higher information density*. For instance, Llama-3.1-8B achieves 53.33% accuracy with a 14.75% ratio in the highest-scoring group, compared to 44.20% with only 8.58% in the lowest. (2) *Longer*

| Model | PiCSAR Rank | Avg Peak Count | Avg Sentences | Avg Peak-to-Sentence Ratio | Accuracy |
|---|---|---|---|---|---|
| Llama-3.1-8B | Highest | 1.88 | 16.43 | 14.75% | 53.33% |
| | Middle (Third Ranked) | 2.00 | 22.86 | 12.75% | 48.80% |
| | Lowest | 2.47 | 64.72 | 8.58% | 44.20% |
| Llama-3.1-70B | Highest | 1.80 | 14.09 | 15.53% | 63.67% |
| | Middle (Third Ranked) | 1.83 | 19.87 | 12.98% | 60.40% |
| | Lowest | 3.08 | 38.37 | 10.83% | 59.40% |
| Qwen3-8B | Highest | 1.99 | 15.78 | 17.63% | 73.67% |
| | Middle (Third Ranked) | 1.91 | 17.57 | 16.95% | 72.80% |
| | Lowest | 2.18 | 26.39 | 14.19% | 69.40% |
| Qwen3-32B | Highest | 1.48 | 11.62 | 22.39% | 77.00% |
| | Middle (Third Ranked) | 1.57 | 12.02 | 19.43% | 76.80% |
| | Lowest | 1.76 | 25.12 | 16.11% | 72.60% |
| Gemma-2-9B | Highest | 1.46 | 8.50 | 24.52% | 46.53% |
| | Middle (Third Ranked) | 1.38 | 9.98 | 18.99% | 44.00% |
| | Lowest | 1.20 | 11.58 | 14.32% | 41.60% |

Table 3: Peak count analysis across different PiCSAR confidence rankings. *We observe that reasoning chains that lead to the correct answer tend to have a higher peak-to-sentence-ratio.*

*reasoning chains do not necessarily improve accuracy.* Table 3 shows that the lowest-ranked responses are substantially longer yet less accurate. For example, Llama-3.1-8B averages 64.72 sentences with 44.20% accuracy in the lowest group, versus 16.43 sentences with 53.33% accuracy in the highest group. This observation aligns with recent findings of inverse scaling in test-time compute (Chen et al., 2024; Wu et al., 2025; Hassid et al., 2025; Ghosal et al., 2025; Gema et al., 2025), showing that solely extended reasoning length does not guarantee improved performance.

## 5.2 DUALITY OF CONFIDENCE: INTRA-MODEL RELIABILITY VS. INTER-MODEL VARIANCE

In this section, we investigate the reliability of PiCSAR for predicting correctness within individual models (*intra-model reliability analysis*) and examine whether these confidence scores remain comparable across different models (*inter-model variance analysis*). For the *intra-model reliability analysis*, we fit regressions for the Qwen and Llama families (Figure 5), with correctness (correct/incorrect) as the dependent variable and the answer confidence score as the independent variable. This approach allows us to interpret the regression slope ($\beta$), which represents the incremental change in log-odds of correctness per unit increase in confidence score.

We find that the $\beta$ is consistently positive across all model sizes, indicating a strong positive relationship between a sample's confidence score and its likelihood of being correct. For example, Qwen3-14B shows a $\beta$ of 0.7255, implying that for every unit increase in the log-probability score, the odds of the answer being correct increase by a factor of over two ($e^{0.7255} \approx 2.07$). The Point-Biserial Correlation Coefficient further supports the positive rela-

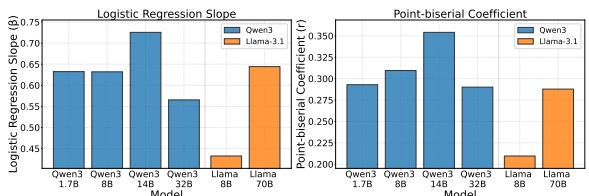

Figure 5: Calibration summary for Qwen3 and Llama-3.1-8B models. We show that the $\beta$ and r coefficients are consistently positive across all models.

tionship by measuring the linear association between binary correctness and continuous confidence scores. *These findings confirm that PiCSAR serves as a reliable predictor of correctness within individual models.* Details of both methods are in Appendix F.

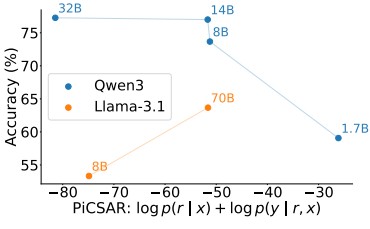

Figure 6: Comparison of % and PiCSAR score.

However, *inter-model variance analysis* challenges the assumption that confidence scores represent universal correctness measures across different models. While intra-model reliability remains stable across different model sizes and architectures, confidence scores cannot be compared across models of different parameter sizes and architectures. As shown in Figure 6, the Llama family exhibits predictable trend: both accuracy and confidence increase with model size. In contrast, the Qwen family shows a non-monotonic relationship; Qwen3-1.7B achieves the highest confidence while showing the lowest accuracy. *This difference implies that while there is a general*

*trend that confidence is a useful proxy for selecting an accurate reasoning path from a set of candidates within models, but its actual value is model-specific and incomparable across different models.*

## 5.3 CONFIDENCE PORTABILITY: DECOUPLING GENERATION FROM EVALUATION

Having established the properties of the confidence signal within a single model, we extend our analysis to multi-model scenarios, evaluating confidence signal robustness when generation and evaluation are decoupled. This decoupling is motivated by practical system design, where one might use a costly API model for reasoning confidence, while relying on a smaller local model for answer confidence estimation. In this *decoupled* setting, the model that generates the reasoning chain ($M_{\text{gen}}$) differs from the model that evaluates the answer confidence ($M_{\text{eval}}$). The scoring function for a chain $r_i$ generated by $M_{\text{gen}}$ becomes:

$$\text{Score}(r_i, y_i) = \underbrace{\log p(r_i \mid x; M_{\text{gen}})}_{\text{Generated by } M_{\text{gen}}} + \underbrace{\log p(y_i \mid \langle a \rangle, r_i, x; M_{\text{eval}})}_{\text{Evaluated by } M_{\text{eval}}}. \quad (4)$$

We test this by having $M_{\text{gen}}$ generate reasoning chains, and various models acting as $M_{\text{eval}}$. For LRMs, the base instruct model is used as $M_{\text{eval}}$.

Our results, detailed in Figure 7 and Appendix A, demonstrate that overall accuracy remains largely unaffected under this decoupling, with only minor degradation even when $M_{\text{eval}}$ is a significantly smaller model than $M_{\text{gen}}$. For instance, accuracy remains similar when $M_{\text{gen}}$ is generated by Llama-3.1-70B, while $M_{\text{eval}}$ is estimated with either Llama-3.1-8B, or other smaller models. This suggests that the answer confidence term, $\log p(y \mid r, x)$, is not merely a model-specific artefact but functions as a more portable measure of the logical entailment between a given reasoning chain and its conclusion. This property enables flexible and computationally efficient answer confidence prediction.

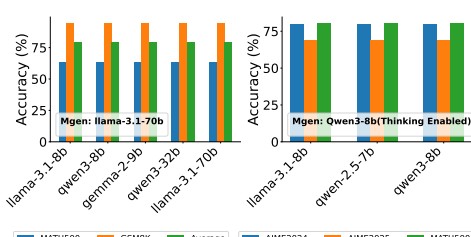

Figure 7: Decoupling analysis for Llama-3.1-70B and Qwen3-8B (Thinking Enabled) as $M_{\text{gen}}$, with various $M_{\text{eval}}$, showing performance remains similar when different models are used to estimate $\log p(y \mid r, x)$.

## 6 RELATED WORK

**Reasoning in LLMs.** Enhancing reasoning abilities of LLMs has yielded significant gains on complex tasks (Li et al., 2025; Muennighoff et al., 2025). While CoT reasoning improves performance (Wei et al., 2022; Leang et al., 2024), subsequent work has introduced hierarchical reasoning phases, including multi-path exploration (Yao et al., 2023; Guan et al., 2025), step verification (Lightman et al., 2024; Leang et al., 2025), and iterative refinement (Madaan et al., 2023). These techniques do not apply to LRMs (Team et al., 2025; Yang et al., 2025a), which typically produce long, unstructured outputs, making the approaches infeasible and computationally expensive.

**Best-of-N (BoN) and Self-Consistency (SC).** BoN is a simple alignment-via-inference method that optimises outputs using a scoring function (Charniak & Johnson, 2005; Stiennon et al., 2020; Amini et al., 2024). Inspired by scale-time inference, LLMs benefit from generating multiple samples and selecting the best using reward models (Snell et al., 2024; Wu et al., 2024). Due to the cost of training reward models, training-free alternatives such as Self-Consistency and its variants (Wan et al., 2024; Wang et al., 2023b; Taubenfeld et al., 2025; Lyu et al., 2025) are widely adopted.

**Sampling and Reranking in LLMs.** Re-ranking is another common method to enhance generation quality (Adiwardana et al., 2020; Shen et al., 2021), often involving a trained "verifier" to re-rank candidate solutions, which improves performance on tasks beyond fine-tuning (Cobbe et al., 2021; Guan et al., 2025). Confidence estimation for re-ranking has been explored via sample agreement (Kuhn et al., 2023; Manakul et al., 2023; Tian et al., 2024), via KL Divergence (Kang et al., 2025) or prompting models to verbalise their confidence (Tian et al., 2023; Kadavath et al., 2022).

## 7 CONCLUSION

We introduced PiCSAR, a sample-efficient, training-free scoring function for BoN sampling that selects a reasoning chain by maximising a score decomposed into reasoning confidence and answer confidence. PiCSAR yields consistent improvements across models and datasets, thereby narrowing the gap to oracle performance. PiCSAR is also sample-efficient, requiring only $k = 6$ samples to outperform baselines using $k = 32$ samples. The answer confidence component can be estimated by different models than the one used for generation, enabling flexible and computationally efficient deployment. At the trajectory level, peak-count-to-sentence ratios correlate with accuracy, showing that reasoning chains leading to correct answers are more information-dense. However, while confidence is predictive within a model, its absolute values remain model-specific and cannot rank models. Overall, PiCSAR offers a promising probabilistic confidence route to reasoning selection.

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

## A  ADDITIONAL RESULTS FOR DECOUPLED CONFIDENCE ESTIMATION

In this section, we provide supplementary evidence that the decoupled confidence estimation experiments introduced in Section 5.3 are portable across distinct evaluator models. This analysis aims to strengthen the claim that the answer-confidence term, $\log p(y \mid r, x)$, does not depend on the specific evaluator used.

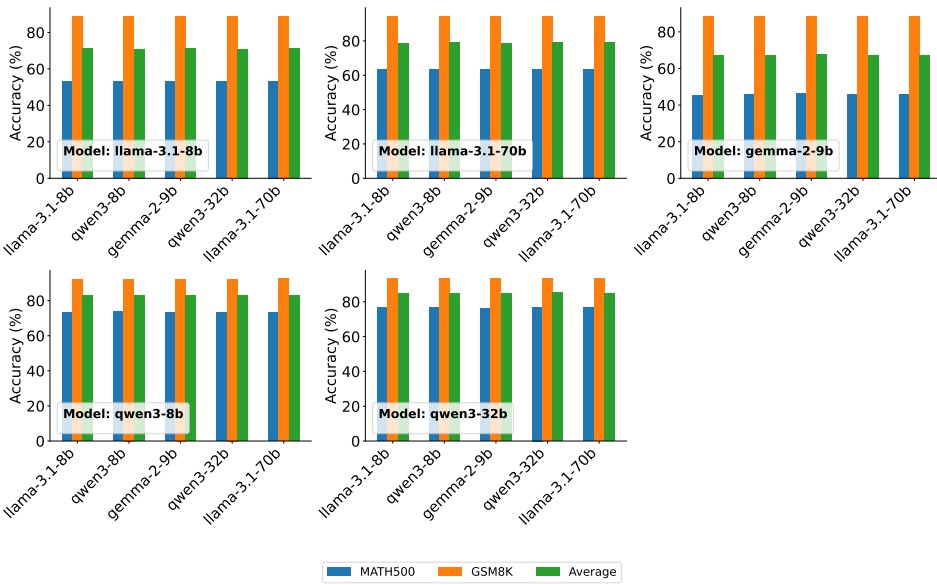

Figure 8: Decoupling plot by using various LLMs to evaluate $p(y \mid r, x)$ across a particular model reasoning chain, $p(r \mid x)$. Each subplot represents a $M_{gen}$, and the *x-axis* represents various $M_{eval}$. The results remain similar when $M_{eval}$ varies, even with smaller models predicting larger $M_{gen}$.

Based on Figure 8, switching the evaluator model, $M_{eval}$ while holding the reasoning distribution fixed yields a similar accuracy across datasets. This observation shows that the answer-confidence term, $\log p(y \mid r, x)$, is highly portable, allowing small-scale LLMs to reliably evaluate the reasoning chains of larger models.

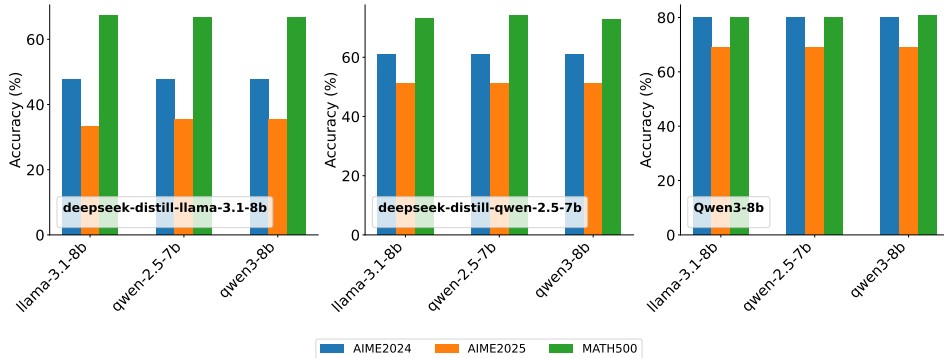

Figure 9: Decoupling plot by using various LLMs to evaluate $p(y \mid r, x)$ across a particular model reasoning chain, $p(r \mid x)$. Each subplot represents a $M_{gen}$, and the *x-axis* represents various $M_{eval}$. The results remain similar when $M_{eval}$ varies, even with smaller models predicting larger $M_{gen}$.

When examining LRMs, we observe the same qualitative pattern (shown in Figure 9), indicating that the phenomenon generalises across models. This reinforces the hypothesis that decoupled confidence estimation captures a stable property of the reasoning process itself, rather than an artefact of the evaluator model.

# B   ADDITIONAL IMPLEMENTATION DETAILS

**Sampling and Decoding.**   For sampling-based methods, we use $k \in \{6, 32\}$ reasoning traces for smaller models and $k \in \{6, 16\}$ for the larger Llama-3.1-70B and Qwen3-32B models, due to computational constraints. For all the models, we apply a hyperparameter of temperature=0.7 and top-p=0.6. The greedy decoding (temperature=0, top-p=1.0) baseline corresponds to $k = 1$, for which we report Pass@1 accuracy. For specialised LRMs, we use $k = 6$ uniformly across all methods due to computational constraints. Since LRMs are not typically evaluated using greedy decoding, we follow the approach of Yang et al. (2025a), which is a temperature of 0.6, top-k of 20 and top-p=0.95, reporting the average accuracy across $k$ samples. For all our baselines except greedy decoding, we evaluate three times with the standard error reported. For LLMs, we cap the maximum token budget at 8,096 tokens. For LRMs, we follow the configuration of Yang et al. (2025a), using a maximum output length of 32,768 tokens, except for AIME'24 and AIME'25, where we extend the budget to 38,912 tokens to ensure sufficient reasoning space.

**Baselines and Hyperparameters**   We compare PiCSAR against a range of decoding, confidence and re-ranking baselines.

- **Greedy Decoding** As a deterministic decoding strategy, greedy decoding selects at each step the token with the highest conditional probability. Unlike greedy decoding, which selects a single high-probability continuation, PiCSAR evaluates multiple full reasoning trajectories and ranks them using joint reasoning-and-answer log-likelihood, enabling selection of the most globally probable chain.
- **Self-Consistency (SC)** (Wang et al., 2023b). This method samples $k$ reasoning chains and aggregates predictions via majority voting on the final answer. In cases where multiple answers receive equal support, we break ties by selecting one at random. While SC relies purely on majority voting over final answers, PiCSAR incorporates the full reasoning chain's token-level likelihood along with answer confidence, allowing it to prefer coherent but minority reasoning paths that SC would discard.
- **Universal Self-Consistency (USC)** (Chen et al., 2023b). We include USC only for LLMs under $k$=6 sampling, as prompt and context length restrictions prevent its application in the LRM setting. We use the prompting strategy proposed in Chen et al. (2023b). Unlike USC, which asks the model to internally judge "consistency" among samples, PiCSAR uses a probabilistic, model-agnostic scoring function based directly on log-likelihoods of reasoning and answers, avoiding USC's reliance on model self-evaluation and context-window limits.
- **Self-Certainty** (Kang et al., 2025). This method applies KL-divergence-based confidence scores, aggregated via Borda voting with parameter $p$=0.5. It provides a probabilistic variant of self-consistency, where each candidate's confidence distribution informs the re-ranking process. Instead of re-ranking chains with KL-based self-estimated correctness like Self-Certainty, PiCSAR scores each candidate through the true generative probabilities of its entire reasoning path and answer
- *P(True)* (Kadavath et al., 2022). This method prompts the model to evaluate whether the answer or reasoning is *True* or *False*, then parses the probability of the response. While P(True) extracts a scalar correctness probability from a meta-prompt, PiCSAR leverages the actual likelihood structure of the model's forward pass, combining reasoning and answer probabilities without relying on verbalized or poorly calibrated self-judgments.
- **CISC** (Taubenfeld et al., 2025). This method aggregates multiple sampled reasoning paths by weighting each path's vote with the model's own estimated correctness. For a fair comparison, we compare CISC with PiCSAR as estimated correctness, termed CISC (PiCSAR), with CISC (*P(True)*), which originally proposed, in Appendix C.1.

We have summarised the novelty of PiCSAR against other baselines in table 4.

**Baseline Restrictions**   Due to context length constraints, USC can only handle a limited number of samples and is therefore evaluated exclusively in the LLM setting with $k$=6, and excluded from all LRM experiments.

**Ablations**   To disentangle the contributions of the two terms in our joint objective, we introduce single-term ablations. *Reasoning Confidence* ranks candidates solely by $\log p(r \mid x)$, favouring

| Method | SC | USC | Self-Cert. | PiCSAR |
|---|---|---|---|---|
| Full Reasoning Chain | ✓ | ✓ | | ✓ |
| Model Confidence | | ✓ | ✓ | ✓ |
| Computationally Efficient | ✓ | ×* | ✓ | ✓ |
| Smaller Model Capable | ✓ | | ✓ | ✓ |

*Due to context length

Table 4: Comparison of Different Methods

plausible reasoning traces. *Answer Confidence* instead ranks by $\log p(y \mid r, x)$, prioritising certainty in the final answer given the reasoning path.

**Framework and Hardware.**   All experiments are conducted using the vLLM framework (Kwon et al., 2023). All experiments are conducted on 2–4 NVIDIA H100 GPUs (80GB). Results are reported as averages over independent evaluation runs to ensure robustness.

**Prompt**   For the reasoning confidence $\log p(r \mid x)$ generation, we utilise the following prompt:

```
You are a helpful AI Assistant that provides well-reasoned and
detailed responses.  Think step by step and provide the final
answer in the form of 'The final answer is:  [answer]'.  Decompose
and break down your reasoning into smallest possible steps (Do
not combine multiple inferences in one step), and do label your
steps very clearly with 'Step 1...  \n\n Step 2...  \n\n Step 3....
\n\n.....  \n\n Step N-1.....  \n\n Step N \n\n The final answer is:
[answer]'.
```

For predicting answer confidence $\log p(y \mid r, x)$, we follow a similar method to (Ton et al., 2024) but without training. Specifically, we use the prompt template $\langle a \rangle$ with 5-shot learning:

```
You are a helpful assistant.  When you see a potential partial
reasoning followed by '<sep>', output the final answer.
```

### B.1   ANALYIS OF PROMPTS

To verify that the observed improvements are not attributable to the explicit instruction prompt (see Equation 3), we evaluated several alternative prompt formulations on the Llama-3.1-8B model. Using the MATH500 benchmark, we compared the resulting answer-confidence estimates across prompts.

```
Prompt 1:  "You are a helpful assistant.  When you see a potential
partial reasoning followed by '<sep>', output the final answer.
Here are some examples" + system_contents + "You are not allowed to
provide any redundant symbols at for the final answer, including
'#', '/', '$', '**' or others.  Please only provide numbers as the
final answer."
```

```
Prompt 2 (original prompt):  "You are a helpful assistant.  When you
see a potential partial reasoning followed by '<sep>', output the
final answer.  Here are some examples"
```

```
Prompt 3:  "You are a helpful assistant.  By providing the partial
reasoning, output the final answer directly without any additional
texts."
```

```
Prompt 4:  "You are a helpful assistant.  Based on the reasoning
provided, output the final answer directly without any additional
texts.  Only Provide the final answer."
```

```
Prompt 5:  "You are a helpful assistant.  Provide the final answer
directly without any additional texts (only the final answer) based
on the partial reasoning."
```

| Prompt | Accuracy |
|--------|----------|
| Prompt 1 | 54.60% |
| Prompt 2 | 54.00% |
| Prompt 3 | 54.20% |
| Prompt 4 | 54.40% |
| Prompt 5 | 54.40% |

Table 5: Performance of PiCSAR on Llama-3.1-8B on MATH500 with Different Prompts for Answer-Confidence Extraction

Our results show that changes in prompt phrasing have minimal influence on model performance. This suggests that, although the instructional content of a prompt remains essential for eliciting the final answer, the precise wording plays only a limited role in shaping the model's behaviour.

## C    FURTHER EXPERIMENTAL RESULTS AND ABLATION STUDIES

### C.1    COMPARISON BETWEEN CISC (P(TRUE)) AND CISC (PICSAR)

Based on Table 6, PiCSAR shows a great performance when integrated with weightage voting on CISC (Taubenfeld et al., 2025), consistently improving baseline CICS (*p(True)*) metrics across all evaluated methods. This indicates that PiCSAR functions effectively both as a standalone selection mechanism and as an augmentation to existing weighting schemes. While these findings suggest promising direction for performance optimisation, this lies beyond the current research scope.

### C.2    COMPONENT ANALYSIS AND MAIN RESULTS BREAKDOWN

In this section, we first provide a detailed breakdown of the experimental results for all methods, as summarised in Table 7, and then we introduce and analyse the performance of PiCSAR-N, a length-normalised variant of our primary method. Finally, we present ablation studies on LRMs in Table 8. We compare three primary approaches: *Reasoning Confidence* ($\log p(r \mid x)$), *Answer Confidence* ($\log p(y \mid r, x)$), and our main method, *PiCSAR* (the joint probability).

Across the majority of benchmarks and model families presented in Table 7, we generally observe that PiCSAR outperforms its individual components. This pattern underscores the benefit of jointly considering the likelihood of both the reasoning process and the final answer. However, there are specific instances where relying solely on answer confidence, $\log p(y \mid r, x)$, achieves comparable or slightly better results (e.g., Gemma-2-9B and Qwen3-32B on GPQA-Diamond for $k = 32$), highlighting that answer confidence remains a strong and competitive signal on its own.

### C.3    ANALYSIS OF LENGTH-NORMALISED VARIANT: PICSAR-N

As introduced in the main paper, we proposed a variant of our method, PiCSAR-N, which applies length normalisation to the reasoning confidence term. The scoring function for PiCSAR-N is defined as:

$$\text{Score}(r, y) = \left[\frac{1}{N} \log p(r \mid x)\right] + \log p(y \mid \langle a \rangle, r, x), \tag{5}$$

| Method | SVAMP | | GSM8K | | MATH500 | | TheoremQA | |
|---|---|---|---|---|---|---|---|---|
| | $k = 6$ | $k = 16/32$ | $k = 6$ | $k = 16/32$ | $k = 6$ | $k = 16/32$ | $k = 6$ | $k = 16/32$ |
| *Gemma-2-9B-Instruct* | | | | | | | | |
| CISC (*p(True)*) | $89.22_{\pm0.22}$ | $88.67_{\pm0.38}$ | $88.89_{\pm0.26}$ | $89.14_{\pm0.15}$ | $46.87_{\pm0.33}$ | $47.67_{\pm0.07}$ | $17.09_{\pm0.43}$ | $17.45_{\pm0.12}$ |
| PiCSAR | $89.00_{\pm0.38}$ | $91.02_{\pm0.59}$ | $88.66_{\pm0.11}$ | $88.99_{\pm0.20}$ | $46.53_{\pm0.29}$ | $47.13_{\pm0.13}$ | $18.62_{\pm0.39}$ | $18.88_{\pm0.54}$ |
| CISC (PiCSAR) | $\mathbf{91.89_{\pm0.22}}$ | $\mathbf{92.33_{\pm0.19}}$ | $\mathbf{91.85_{\pm0.20}}$ | $\mathbf{92.43_{\pm0.22}}$ | $\mathbf{51.33_{\pm0.07}}$ | $\mathbf{52.13_{\pm0.29}}$ | $\mathbf{21.02_{\pm0.58}}$ | $\mathbf{23.16_{\pm0.39}}$ |
| Upper Bound | $24.32_{\pm0.49}$ | $32.40_{\pm0.20}$ | $93.44_{\pm0.09}$ | $95.60_{\pm0.04}$ | $58.47_{\pm0.27}$ | $66.67_{\pm0.47}$ | $55.22_{\pm1.10}$ | $82.49_{\pm1.02}$ |
| *Llama-3.1-8B-Instruct* | | | | | | | | |
| CISC (*p(True)*)) | $91.44_{\pm0.48}$ | $92.78_{\pm0.29}$ | $91.17_{\pm0.18}$ | $91.91_{\pm0.49}$ | $54.93_{\pm0.41}$ | $58.20_{\pm0.42}$ | $18.03_{\pm0.73}$ | $39.38_{\pm18.91}$ |
| PiCSAR | $91.78_{\pm0.11}$ | $93.44_{\pm0.89}$ | $89.09_{\pm0.13}$ | $89.98_{\pm0.23}$ | $53.33_{\pm0.73}$ | $53.87_{\pm0.70}$ | $20.08_{\pm0.43}$ | $19.72_{\pm0.39}$ |
| CISC (PiCSAR) | $\mathbf{94.33_{\pm0.33}}$ | $\mathbf{96.22_{\pm0.11}}$ | $\mathbf{93.98_{\pm0.14}}$ | $\mathbf{94.23_{\pm0.08}}$ | $\mathbf{62.47_{\pm0.07}}$ | $\mathbf{62.40_{\pm0.50}}$ | $\mathbf{22.71_{\pm0.25}}$ | $\mathbf{41.50_{\pm17.34}}$ |
| Upper Bound | $96.78_{\pm0.11}$ | $99.11_{\pm0.11}$ | $96.15_{\pm0.07}$ | $98.18_{\pm0.04}$ | $72.80_{\pm0.23}$ | $82.20_{\pm0.60}$ | $28.20_{\pm0.32}$ | $37.846_{\pm1.13}$ |
| *Qwen3-8B (Non-thinking)* | | | | | | | | |
| CICS (*p(True)*)) | $94.33_{\pm0.00}$ | $94.56_{\pm0.11}$ | $93.80_{\pm0.13}$ | $94.05_{\pm0.14}$ | $77.20_{\pm0.20}$ | $77.93_{\pm0.24}$ | $31.24_{\pm0.04}$ | $32.75_{\pm0.45}$ |
| PiCSAR | $93.56_{\pm0.22}$ | $95.13_{\pm0.22}$ | $92.33_{\pm0.13}$ | $93.22_{\pm0.08}$ | $73.67_{\pm0.24}$ | $73.40_{\pm0.13}$ | $29.76_{\pm0.57}$ | $29.17_{\pm0.64}$ |
| CICS (PiCSAR) | $\mathbf{95.11_{\pm0.11}}$ | $\mathbf{95.67_{\pm0.19}}$ | $\mathbf{94.89_{\pm0.14}}$ | $\mathbf{95.22_{\pm0.12}}$ | $\mathbf{79.80_{\pm0.40}}$ | $\mathbf{79.60_{\pm0.42}}$ | $\mathbf{36.46_{\pm0.04}}$ | $\mathbf{36.32_{\pm0.04}}$ |
| Upper Bound | $96.33_{\pm0.67}$ | $97.89_{\pm0.11}$ | $95.52_{\pm0.00}$ | $96.84_{\pm0.03}$ | $81.13_{\pm0.44}$ | $83.53_{\pm0.24}$ | $34.94_{\pm0.00}$ | $40.03_{\pm0.35}$ |
| *Llama-3.1-70B-Instruct* | | | | | | | | |
| CISC (*p(True)*)) | $94.22_{\pm0.22}$ | $94.11_{\pm0.11}$ | $94.68_{\pm0.00}$ | $95.09_{\pm0.09}$ | $65.07_{\pm1.05}$ | $66.27_{\pm0.29}$ | $28.07_{\pm0.68}$ | $29.41_{\pm0.12}$ |
| PiCSAR | $94.10_{\pm0.11}$ | $95.58_{\pm0.22}$ | $94.58_{\pm0.03}$ | $94.81_{\pm0.13}$ | $63.67_{\pm1.51}$ | $64.07_{\pm0.87}$ | $27.84_{\pm0.19}$ | $26.73_{\pm0.27}$ |
| CISC (PiCSAR) | $\mathbf{96.78_{\pm0.11}}$ | $\mathbf{96.44_{\pm0.11}}$ | $\mathbf{95.90_{\pm0.08}}$ | $\mathbf{96.03_{\pm0.11}}$ | $\mathbf{69.60_{\pm0.31}}$ | $\mathbf{70.80_{\pm0.76}}$ | $\mathbf{31.91_{\pm0.31}}$ | $\mathbf{31.59_{\pm0.27}}$ |
| Upper Bound | $97.22_{\pm0.22}$ | $97.78_{\pm0.22}$ | $96.91_{\pm0.03}$ | $97.44_{\pm0.03}$ | $77.07_{\pm0.47}$ | $81.67_{\pm0.18}$ | $40.70_{\pm0.20}$ | $43.47_{\pm0.18}$ |
| *Qwen3-32B (Non-thinking)* | | | | | | | | |
| CICS (P-True) | $94.33_{\pm0.00}$ | $94.56_{\pm0.11}$ | $93.80_{\pm0.13}$ | $94.05_{\pm0.14}$ | $77.20_{\pm0.20}$ | $77.93_{\pm0.24}$ | $31.24_{\pm0.04}$ | $32.75_{\pm0.45}$ |
| PiCSAR | $93.22_{\pm0.22}$ | $93.55_{\pm0.33}$ | $93.90_{\pm0.28}$ | $93.88_{\pm0.22}$ | $77.00_{\pm0.18}$ | $75.93_{\pm0.13}$ | $31.46_{\pm0.04}$ | $31.42_{\pm0.27}$ |
| CICS (PiCSAR) | $\mathbf{95.11_{\pm0.11}}$ | $\mathbf{95.67_{\pm0.19}}$ | $\mathbf{94.89_{\pm0.14}}$ | $\mathbf{95.22_{\pm0.12}}$ | $\mathbf{79.80_{\pm0.40}}$ | $\mathbf{79.60_{\pm0.42}}$ | $\mathbf{36.46_{\pm0.04}}$ | $\mathbf{36.32_{\pm0.04}}$ |
| Upper Bound | $96.78_{\pm0.11}$ | $98.00_{\pm0.00}$ | $96.28_{\pm0.13}$ | $96.99_{\pm0.07}$ | $82.27_{\pm0.13}$ | $83.73_{\pm0.07}$ | $39.76_{\pm0.00}$ | $42.93_{\pm0.12}$ |

Table 6: **Performance comparison on benchmarks across CISC (*p(True)*) and CISC (PiCSAR) on LLMs.** Values represent mean accuracy ± standard error over three independent evaluation runs. **Bold** indicates the best-performing method per column based on the mean accuracy. Sampling parameters: $k = \{6, 32\}$ for Gemma-2-9B, Llama-3.1-8B, and Qwen3-8B; $k = \{6, 16\}$ for Llama-3.1-70B and Qwen3-32B.

where $N$ is the number of tokens in the reasoning chain $r$. This normalisation is intended to mitigate any potential bias against longer, more detailed reasoning paths which might be unfairly penalised by the sum of negative log-probabilities.

## C.4 ANALYSIS BETWEEN TOKEN LENGTH, PiCSAR SCORE, AND MODEL PERFORMANCE

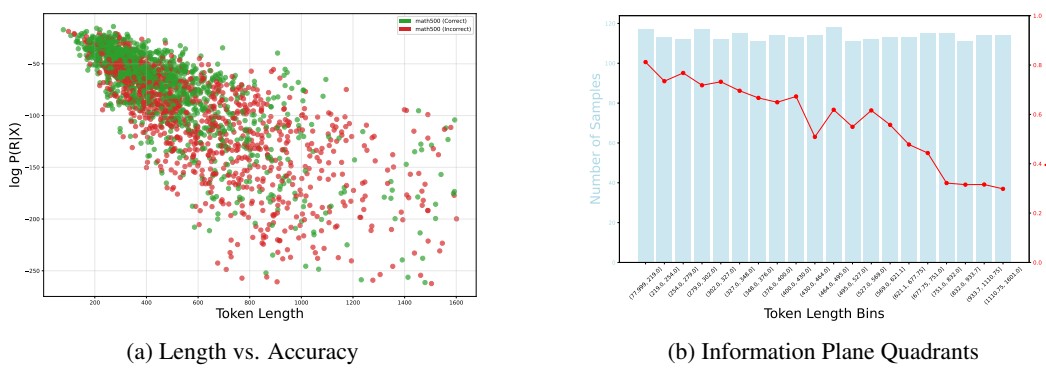

(a) Length vs. Accuracy

(b) Information Plane Quadrants

Figure 10: Relationship between token length, probability, and accuracy.

Figure 10a shows that correct instances predominantly cluster in regions of high probability and short sequence length, indicating that concise reasoning is strongly associated with higher quality. This pattern is reinforced by Figure 10b, which demonstrates a consistent decline in accuracy as sequence length grows. Together, the two figures highlight that shorter, more confident reasoning trajectories tend to yield more accurate performance.

| Method | SVAMP | | GSM8K | | MATH500 | | GPQA-Diamond | |
|---|---|---|---|---|---|---|---|---|
| | $k=6$ | $k=16/32$ | $k=6$ | $k=16/32$ | $k=6$ | $k=16/32$ | $k=6$ | $k=16/32$ |
| *Gemma-2-9B-Instruct* | | | | | | | | |
| Reasoning Confidence | $88.66_{\pm0.33}$ | $89.67_{\pm0.49}$ | $88.51_{\pm0.05}$ | $88.46_{\pm0.25}$ | $45.87_{\pm0.47}$ | $45.87_{\pm0.68}$ | $30.64_{\pm0.45}$ | $32.32_{\pm1.52}$ |
| Answer Confidence | $89.66_{\pm0.33}$ | $89.02_{\pm0.59}$ | $88.05_{\pm0.17}$ | $87.04_{\pm0.05}$ | $46.47_{\pm0.66}$ | $46.33_{\pm0.18}$ | $34.01_{\pm2.65}$ | $\mathbf{38.22}_{\pm1.76}$ |
| Reasoning confidence (normalised) | $89.56_{\pm0.44}$ | $90.22_{\pm0.59}$ | $88.76_{\pm0.26}$ | $\mathbf{89.45}_{\pm0.20}$ | $46.33_{\pm0.67}$ | $46.47_{\pm0.18}$ | $29.80_{\pm1.91}$ | $27.95_{\pm2.15}$ |
| PiCSAR | $89.00_{\pm0.38}$ | $\mathbf{91.02}_{\pm0.59}$ | $88.66_{\pm0.11}$ | $88.99_{\pm0.20}$ | $46.53_{\pm0.29}$ | $\mathbf{47.13}_{\pm0.13}$ | $32.32_{\pm0.51}$ | $34.01_{\pm1.94}$ |
| PiCSAR-N | $\mathbf{89.67}_{\pm0.19}$ | $89.22_{\pm0.29}$ | $\mathbf{88.91}_{\pm0.12}$ | $89.27_{\pm0.11}$ | $\mathbf{46.60}_{\pm0.92}$ | $46.93_{\pm0.18}$ | $\mathbf{35.35}_{\pm1.62}$ | $38.05_{\pm1.90}$ |
| Upper Bound | $93.44_{\pm0.22}$ | $95.67_{\pm0.38}$ | $93.44_{\pm0.09}$ | $95.60_{\pm0.04}$ | $58.47_{\pm0.27}$ | $66.67_{\pm0.47}$ | $55.22_{\pm1.10}$ | $82.49_{\pm1.02}$ |
| *Llama-3.1-8B-Instruct* | | | | | | | | |
| Reasoning Confidence | $91.56_{\pm0.11}$ | $92.10_{\pm0.84}$ | $88.89_{\pm0.09}$ | $89.67_{\pm0.27}$ | $53.07_{\pm0.37}$ | $51.53_{\pm0.35}$ | $29.12_{\pm1.02}$ | $32.49_{\pm2.92}$ |
| Answer Confidence | $89.11_{\pm0.29}$ | $90.44_{\pm0.95}$ | $86.84_{\pm0.20}$ | $86.69_{\pm0.04}$ | $49.27_{\pm0.64}$ | $50.20_{\pm0.35}$ | $28.62_{\pm0.73}$ | $29.46_{\pm2.63}$ |
| Reasoning confidence (normalised) | $90.22_{\pm0.11}$ | $90.67_{\pm0.69}$ | $88.38_{\pm0.23}$ | $86.10_{\pm0.08}$ | $50.67_{\pm0.47}$ | $47.13_{\pm1.39}$ | $22.05_{\pm0.89}$ | $18.35_{\pm0.84}$ |
| PiCSAR | $\mathbf{91.78}_{\pm0.11}$ | $\mathbf{93.44}_{\pm0.89}$ | $\mathbf{89.09}_{\pm0.13}$ | $\mathbf{89.98}_{\pm0.23}$ | $\mathbf{53.33}_{\pm0.73}$ | $\mathbf{53.87}_{\pm0.70}$ | $29.80_{\pm1.34}$ | $\mathbf{33.67}_{\pm3.06}$ |
| PiCSAR-N | $90.22_{\pm0.48}$ | $92.22_{\pm0.29}$ | $88.59_{\pm0.18}$ | $89.33_{\pm0.42}$ | $51.53_{\pm0.48}$ | $51.60_{\pm0.42}$ | $\mathbf{30.81}_{\pm0.87}$ | $30.64_{\pm1.61}$ |
| Upper Bound | $96.78_{\pm0.11}$ | $99.11_{\pm0.11}$ | $96.15_{\pm0.07}$ | $98.18_{\pm0.04}$ | $72.80_{\pm0.23}$ | $82.20_{\pm0.60}$ | $65.82_{\pm1.50}$ | $92.76_{\pm0.73}$ |
| *Qwen3-8B (Non-thinking)* | | | | | | | | |
| Reasoning Confidence | $92.78_{\pm0.11}$ | $94.34_{\pm0.33}$ | $92.26_{\pm0.13}$ | $92.31_{\pm0.03}$ | $73.53_{\pm0.24}$ | $72.53_{\pm0.48}$ | $45.96_{\pm1.01}$ | $43.77_{\pm1.21}$ |
| Answer Confidence | $93.45_{\pm0.19}$ | $94.02_{\pm0.40}$ | $93.22_{\pm0.03}$ | $92.94_{\pm0.17}$ | $71.07_{\pm0.41}$ | $71.20_{\pm0.76}$ | $\mathbf{51.01}_{\pm1.52}$ | $43.43_{\pm2.53}$ |
| Reasoning Confidence (normalised) | $93.33_{\pm0.00}$ | $93.67_{\pm0.69}$ | $92.79_{\pm0.00}$ | $92.61_{\pm0.20}$ | $71.93_{\pm0.71}$ | $69.27_{\pm0.44}$ | $43.43_{\pm0.51}$ | $38.05_{\pm1.78}$ |
| PiCSAR | $93.56_{\pm0.22}$ | $\mathbf{95.13}_{\pm0.22}$ | $92.33_{\pm0.13}$ | $93.22_{\pm0.08}$ | $73.67_{\pm0.24}$ | $\mathbf{73.40}_{\pm0.13}$ | $46.98_{\pm1.01}$ | $43.69_{\pm1.26}$ |
| PiCSAR-N | $\mathbf{94.44}_{\pm0.11}$ | $94.56_{\pm0.59}$ | $\mathbf{93.69}_{\pm0.00}$ | $\mathbf{93.77}_{\pm0.13}$ | $\mathbf{73.80}_{\pm0.20}$ | $72.13_{\pm0.98}$ | $47.98_{\pm1.01}$ | $\mathbf{44.95}_{\pm0.58}$ |
| Upper Bound | $96.33_{\pm0.67}$ | $97.89_{\pm0.11}$ | $95.52_{\pm0.00}$ | $96.84_{\pm0.03}$ | $81.13_{\pm0.44}$ | $83.53_{\pm0.24}$ | $76.26_{\pm1.62}$ | $86.36_{\pm0.29}$ |
| *Llama-3.1-70B-Instruct* | | | | | | | | |
| Reasoning Confidence | $94.44_{\pm0.11}$ | $94.80_{\pm0.19}$ | $94.46_{\pm0.08}$ | $93.62_{\pm0.18}$ | $63.47_{\pm1.35}$ | $63.00_{\pm0.10}$ | $43.94_{\pm2.62}$ | $45.96_{\pm2.54}$ |
| Answer Confidence | $93.89_{\pm0.22}$ | $94.67_{\pm0.38}$ | $94.10_{\pm0.25}$ | $94.68_{\pm0.23}$ | $59.40_{\pm1.30}$ | $60.07_{\pm1.09}$ | $45.12_{\pm0.45}$ | $42.26_{\pm1.78}$ |
| Reasoning Confidence (normalised) | $93.33_{\pm0.38}$ | $93.89_{\pm0.22}$ | $93.37_{\pm0.03}$ | $93.34_{\pm0.26}$ | $65.60_{\pm0.60}$ | $65.13_{\pm0.13}$ | $40.07_{\pm1.87}$ | $37.04_{\pm0.89}$ |
| PiCSAR | $94.10_{\pm0.11}$ | $\mathbf{95.58}_{\pm0.22}$ | $\mathbf{94.58}_{\pm0.03}$ | $\mathbf{94.81}_{\pm0.13}$ | $63.67_{\pm1.51}$ | $64.07_{\pm0.87}$ | $46.91_{\pm2.65}$ | $\mathbf{46.46}_{\pm2.59}$ |
| PiCSAR-N | $\mathbf{94.44}_{\pm0.11}$ | $94.56_{\pm0.59}$ | $94.07_{\pm0.00}$ | $94.14_{\pm0.13}$ | $\mathbf{72.00}_{\pm0.20}$ | $\mathbf{70.33}_{\pm0.98}$ | $\mathbf{47.98}_{\pm1.01}$ | $44.95_{\pm0.58}$ |
| Upper Bound | $97.22_{\pm0.22}$ | $97.78_{\pm0.22}$ | $96.91_{\pm0.03}$ | $97.44_{\pm0.03}$ | $77.07_{\pm0.47}$ | $81.67_{\pm0.18}$ | $75.59_{\pm0.61}$ | $87.71_{\pm0.45}$ |
| *Qwen3-32B (Non-thinking)* | | | | | | | | |
| Reasoning confidence | $92.78_{\pm0.22}$ | $93.33_{\pm0.29}$ | $93.19_{\pm0.28}$ | $94.54_{\pm0.22}$ | $76.47_{\pm0.07}$ | $75.87_{\pm0.18}$ | $44.78_{\pm0.94}$ | $42.59_{\pm1.02}$ |
| Answer confidence | $92.56_{\pm0.11}$ | $92.22_{\pm0.29}$ | $93.84_{\pm0.05}$ | $93.42_{\pm0.13}$ | $75.40_{\pm0.46}$ | $74.67_{\pm0.18}$ | $\mathbf{51.85}_{\pm0.61}$ | $44.11_{\pm0.94}$ |
| Reasoning Confidence (normalised) | $93.33_{\pm0.19}$ | $94.11_{\pm0.29}$ | $93.39_{\pm0.00}$ | $93.44_{\pm0.30}$ | $75.47_{\pm0.27}$ | $75.53_{\pm0.18}$ | $49.33_{\pm1.18}$ | $37.88_{\pm1.27}$ |
| PiCSAR | $93.22_{\pm0.22}$ | $93.55_{\pm0.33}$ | $93.90_{\pm0.28}$ | $93.88_{\pm0.22}$ | $\mathbf{77.00}_{\pm0.18}$ | $75.93_{\pm0.13}$ | $46.91_{\pm1.02}$ | $\mathbf{44.44}_{\pm2.28}$ |
| PiCSAR-N | $\mathbf{93.33}_{\pm0.38}$ | $\mathbf{93.89}_{\pm0.22}$ | $\mathbf{94.12}_{\pm0.03}$ | $\mathbf{94.09}_{\pm0.26}$ | $76.40_{\pm0.60}$ | $75.13_{\pm0.13}$ | $40.07_{\pm1.87}$ | $37.04_{\pm0.89}$ |
| Upper Bound | $96.78_{\pm0.11}$ | $98.00_{\pm0.00}$ | $96.28_{\pm0.13}$ | $96.99_{\pm0.07}$ | $82.27_{\pm0.13}$ | $83.73_{\pm0.07}$ | $72.56_{\pm1.87}$ | $86.20_{\pm1.02}$ |

Table 7: **Performance comparison on benchmarks across methods on LLMs.** Values represent mean accuracy ± standard error over three independent evaluation runs. **Bold** indicates the best-performing method per column based on the mean accuracy. Sampling parameters: $k = \{6, 32\}$ for Gemma-2-9B, Llama-3.1-8B, and Qwen3-8B; $k = \{6, 16\}$ for Llama-3.1-70B and Qwen3-32B.

## C.5 Ablation Studies on LLMs and LRMs

The results for PiCSAR-N are included in Table 7 and Table 8. As shown, both PiCSAR and PiCSAR-N consistently surpass the other baselines, including their corresponding reasoning confidence metrics (with and without normalisation). The performance difference between PiCSAR and PiCSAR-N is not consistently in one direction; each variant excels on different model-dataset combinations. For instance, PiCSAR-N shows stronger performance with Gemma-2-9B on MATH500 ($k = 6$) and GPQA-Diamond, whereas the non-normalised PiCSAR is clearly superior for Llama-3.1-8B across most settings. This suggests that the utility of length normalisation may depend on model-specific characteristics, such as tendencies towards verbosity.

Based on Table 7, we also observe that 20/40 results of the length-normalised (PiCSAR-N) versions outperform the non-length normalised versions (PiCSAR), demonstrating that length-normalisation does not perform worse than the non-length normalised version. This suggests that length normalisation is not detrimental and does not consistently weaken PiCSAR.

We further conducted ablation studies on LRMs, with results reported in Table 8. Here, we compare PiCSAR and PiCSAR-N against both standard and normalised reasoning confidence, as well as answer confidence. The results confirm that our joint probability methods, PiCSAR and PiCSAR-N, consistently achieve top performance, similar to the findings with LLMs. Interestingly, we observe that maximising answer confidence alone yields strong results, sometimes comparable to PiCSAR, particularly on the DS-Distill-llama-3-8B model. This reinforces the value of the answer confidence signal while highlighting the general effectiveness of PiCSAR's approach in combining both reasoning and answer confidence.

| Method | AIME 2024 | AIME 2025 | MATH500 | SVAMP | GSM8K | GPQA-Diamond |
|---|---|---|---|---|---|---|
| *DS-Distill-llama-3-8B* | | | | | | |
| Reasoning Confidence | $44.43_{\pm5.56}$ | $\mathbf{35.56}_{\pm1.11}$ | $66.60_{\pm0.60}$ | $83.67_{\pm0.00}$ | $72.97_{\pm0.30}$ | $46.97_{\pm0.29}$ |
| Reasoning Confidence (Normalised) | $33.33_{\pm3.85}$ | $28.89_{\pm1.12}$ | $65.70_{\pm1.30}$ | $83.00_{\pm0.13}$ | $76.08_{\pm0.23}$ | $41.41_{\pm1.05}$ |
| Answer Confidence | $42.22_{\pm4.01}$ | $32.22_{\pm1.11}$ | $\mathbf{67.60}_{\pm1.80}$ | $88.33_{\pm0.16}$ | $76.06_{\pm0.43}$ | $\mathbf{48.99}_{\pm1.62}$ |
| PiCSAR | $\mathbf{47.78}_{\pm4.01}$ | $33.33_{\pm1.13}$ | $67.20_{\pm0.60}$ | $85.67_{\pm0.07}$ | $\mathbf{76.42}_{\pm0.16}$ | $47.31_{\pm0.17}$ |
| PiCSAR-N | $40.00_{\pm5.09}$ | $32.22_{\pm1.13}$ | $67.40_{\pm1.00}$ | $\mathbf{89.00}_{\pm0.00}$ | $75.73_{\pm0.41}$ | $47.47_{\pm2.78}$ |
| Upper Bound | $66.67_{\pm5.09}$ | $51.11_{\pm1.11}$ | $82.00_{\pm0.13}$ | $95.67_{\pm0.00}$ | $92.91_{\pm0.35}$ | $77.27_{\pm0.77}$ |
| *DS-Distill-Qwen-2.5-7B* | | | | | | |
| Reasoning Confidence | $57.78_{\pm1.11}$ | $51.11_{\pm1.11}$ | $72.93_{\pm0.81}$ | $91.33_{\pm0.58}$ | $87.83_{\pm0.13}$ | $52.02_{\pm2.81}$ |
| Reasoning Confidence (Normalised) | $54.44_{\pm2.22}$ | $45.56_{\pm2.22}$ | $74.20_{\pm1.10}$ | $90.33_{\pm0.58}$ | $88.26_{\pm0.20}$ | $45.96_{\pm2.67}$ |
| Answer Confidence | $50.00_{\pm5.09}$ | $44.44_{\pm2.22}$ | $72.60_{\pm0.23}$ | $91.00_{\pm0.51}$ | $88.91_{\pm0.08}$ | $\mathbf{53.20}_{\pm2.19}$ |
| PiCSAR | $\mathbf{61.11}_{\pm1.11}$ | $\mathbf{51.11}_{\pm1.11}$ | $\mathbf{74.00}_{\pm0.70}$ | $\mathbf{91.78}_{\pm0.48}$ | $88.18_{\pm0.07}$ | $52.36_{\pm2.88}$ |
| PiCSAR-N | $57.78_{\pm2.22}$ | $48.89_{\pm2.22}$ | $73.40_{\pm1.10}$ | $\mathbf{91.78}_{\pm0.29}$ | $\mathbf{89.60}_{\pm0.18}$ | $50.34_{\pm2.19}$ |
| Upper Bound | $72.22_{\pm1.11}$ | $70.00_{\pm0.00}$ | $83.33_{\pm0.18}$ | $96.33_{\pm0.38}$ | $96.79_{\pm0.13}$ | $79.12_{\pm2.07}$ |
| *Qwen3-8B* | | | | | | |
| Reasoning Confidence | $80.00_{\pm0.00}$ | $68.89_{\pm2.22}$ | $79.20_{\pm0.00}$ | $93.00_{\pm0.33}$ | $95.92_{\pm0.03}$ | $58.59_{\pm1.62}$ |
| Reasoning Confidence (Normalised) | $67.78_{\pm2.22}$ | $65.56_{\pm4.01}$ | $80.00_{\pm0.00}$ | $93.56_{\pm0.56}$ | $95.72_{\pm0.05}$ | $56.23_{\pm1.76}$ |
| Answer Confidence | $76.67_{\pm0.00}$ | $\mathbf{73.33}_{\pm1.92}$ | $80.13_{\pm0.33}$ | $93.78_{\pm0.11}$ | $95.37_{\pm0.00}$ | $60.61_{\pm0.29}$ |
| PiCSAR | $\mathbf{81.33}_{\pm1.34}$ | $68.89_{\pm2.22}$ | $80.60_{\pm0.13}$ | $\mathbf{94.33}_{\pm0.33}$ | $\mathbf{95.94}_{\pm0.04}$ | $59.43_{\pm1.61}$ |
| PiCSAR-N | $76.67_{\pm3.33}$ | $70.00_{\pm5.09}$ | $\mathbf{89.67}_{\pm0.37}$ | $94.22_{\pm0.56}$ | $95.08_{\pm0.03}$ | $\mathbf{61.11}_{\pm1.77}$ |
| Upper Bound | $87.78_{\pm1.11}$ | $82.22_{\pm1.11}$ | $84.00_{\pm0.12}$ | $97.56_{\pm0.11}$ | $97.54_{\pm0.03}$ | $80.13_{\pm0.45}$ |

Table 8: **Performance comparison of model across various baselines and benchmarks on LRMs, measured in terms of accuracy. (%)** For all the evaluations, we use $k = 6$ sampling. *PiCSAR outperforms all baselines with more pronounced gains in more challenging benchmarks.*

## C.6 The Importance of Selection: Interpreting the Upper Bound:

While PiCSAR consistently outperforms other heuristics, it necessarily falls short of the oracle *Upper Bound*, whose behaviour provides insight into the underlying challenges. On easier benchmarks such as SVAMP and GSM8K, the upper bound saturates quickly. For instance, increasing the sample size from $k = 6$ to $k = 32$ with Llama-3.1-70B on GSM8K raises accuracy only marginally from 96.91% to 97.44%, indicating that correct reasoning paths are usually present in small sample sets, and that selection rather than generation is the main bottleneck. In contrast, on more demanding tasks such as MATH500 and GPQA-Diamond, the upper bound continues to rise with larger $k$, as seen with Gemma-2-9B on GPQA-Diamond where accuracy jumps from 55.22% to 82.49%, reflecting the intrinsic difficulty of generating correct answers. In both regimes, PiCSAR demonstrates its value: in selection-limited settings, it reliably identifies correct candidates from small pools, while in generation-limited scenarios, it narrows the gap to the oracle by detecting correct reasoning even when correct answers are sparse, highlighting that improving selection is often as important as enlarging the sampling budget.

## C.7 Analysis of Fallback Mechanism

To assess how sensitive our method is to the penalty assigned when a generation fails, *i.e.,* no answer token is produced and the answer-confidence term cannot be calculated, we tested several fallback values for the Answer Confidence score $(Y)$. Specifically, we compared our default setting of $\log p(y \mid r, X) = -10$ with more conservative penalties of $Y = -20$ and $Y = -100$. As shown in Table 9, downstream accuracy is unchanged across all configurations. This indicates that, as long as the fallback value is sufficiently low to denote a failure state, its precise magnitude does not affect candidate rankings.

| $\log p(y \mid r, X)$ | **Accuracy** |
|---|---|
| $-10$ | 53.40% |
| $-20$ | 53.40% |
| $-100$ | 53.40% |

Table 9: Sensitivity analysis of the Answer Confidence fallback value $(Y)$ on model accuracy. The performance is robust to the magnitude of the penalty.

| Samples | PiCSAR Accuracy | Self-Consistency Accuracy |
|---|---|---|
| 6 | 89.11% | 88.15% |
| 10 | 89.89% | 88.56% |
| 16 | 89.89% | 88.11% |
| 32 | 90.22% | 88.89% |

Table 10: Scaling analysis of GEMMA-2-9B on SVAMP comparing PiCSAR against Self-Consistency across varying sample counts.

### C.8 ANALYSIS OF PERFORMANCE WITH NUMBER OF SAMPLES AND TEMPERATURE

We first examine the scaling behavior of PiCSAR regarding the number of candidate generations ($k$). We evaluate GEMMA-2-9B on the SVAMP dataset with sample budgets ranging from $k = 6$ to $k = 32$. As shown in Table 10, PiCSAR exhibits scaling properties, with accuracy consistently improving as the candidate pool expands (rising from 89.11% at $k = 6$ to 90.22% at $k = 32$). In contrast, Self-Consistency plateaus earlier and remains consistently outperformed by our method. This indicates that PiCSAR is more effective at leveraging larger compute budgets to identify correct reasoning chains.

| Temperature | PiCSAR Accuracy |
|---|---|
| 0.7 | 89.89% |
| 1.0 | 89.67% |

Table 11: Effect of sampling temperature on GEMMA-2-9B performance on SVAMP.

Additionally, we assess the stability of our method with respect to generation stochasticity by comparing performance at sampling temperatures of $T = 0.7$ and $T = 1.0$. The results, summarized in Table 11, reveal negligible performance variance (89.89% vs. 89.67%). These results indicate that PiCSAR is robust to moderate changes in generation hyperparameters and maintains high precision even under more stochastic sampling conditions ($T = 1.0$).

## C.9 EXAMPLE OF PICSAR SCORING

```
Question:  Every day, Wendi feeds each of her chickens three cups
of mixed feed.  She gives the flock three meals per day.  In the
morning, she gives 15 cups of feed, and in the afternoon she gives
another 25 cups.  How many cups of feed must she give in the final
meal of the day if her flock contains 20 chickens?

High PiCSAR Score
Step 1:  Determine the total number of cups of feed given in the
morning.  Wendi gives 15 cups.
Step 2:  Determine the total number of cups of feed given in the
afternoon.  Wendi gives 25 cups.
Step 3:  Total feed given so far = 15 cups + 25 cups = 40 cups.
Step 4:  Each chicken needs 3 cups of feed per day.
Step 5:  Total daily feed required = 3 cups/chicken * 20 chickens =
60 cups.
Step 6:  Total given in morning and afternoon = 40 cups.
Step 7:  Final meal feed = 60 cups - 40 cups = 20 cups.
The final answer is:  20.
PiCSAR Score:  -36.760562883843484

Low PiCSAR Score
Step 1:  Determine the total morning feed.  Wendi gives 15 cups.
Step 2:  Determine the total afternoon feed.  Wendi gives 25 cups.
Step 3:  Total feed in morning and afternoon = 15 + 25 = 40 cups.
Step 4:  Since there are three meals, multiply the morning+afternoon
total by 1.5 to estimate total daily feed.  Total = 40 * 1.5 = 60
cups.
Step 5:  Each chicken is assumed to get 3 cups per meal, so 3 meals
* 3 cups = 9 cups per chicken per day.
Step 6:  Total feed needed per day = 20 chickens * 9 cups = 180
cups.
Step 7:  Final meal feed = 180 cups - 40 cups = 140 cups.
The final answer is:  140.
PiCSAR Score:  -112.79042702552356

We can observe that a concise generation with high probability leads
to higher quality than a low-probability generation with longer
length.
```

## D LIMITATION

PiCSAR primarily targets domains with well-defined reasoning structures and definitive answers, such as mathematical and scientific problem-solving. We view this scope as both deliberate and essential: these domains represent a substantial class of high-value reasoning tasks where precision is paramount. Furthermore, restricting our analysis to these settings enables a rigorous evaluation of confidence calibration, a task that remains notoriously difficult in open-ended domains characterized by ambiguity and multiple valid solutions. This controlled environment allows us to validate the efficacy of model confidence as a selection metric without the confounding factors of subjective evaluation.

Extending PiCSAR to open-ended generation remains an important avenue for future research. To address the lack of definitive answer boundaries in such tasks, a promising direction is to augment the probabilistic framework with learned reward models for answer evaluation. We believe this adaptation could extend the reliability benefits of PiCSAR beyond fixed-format problems, offering a pathway toward robust reasoning in broader, general-purpose applications.

# E ADDITIONAL EXPERIMENTS FOR CONFIDENCE SELECTION METHOD

In this section, we show all the models across datasets (GSM8K, MATH500 and AIME2024), which consist of a variety of difficulties. We observe a consistent pattern across PiCSAR. In addition, the utility of our confidence metric extends to filtering for high-reliability answers. For GSM8K and MATH500, we use the median as our threshold with outliers removed, similar to Section 2.3. However, as for AIME2024, as the instance is similar, we include all the instances include the outliers, and set the threshold to 60% for both x and y-axis.

GSM8K

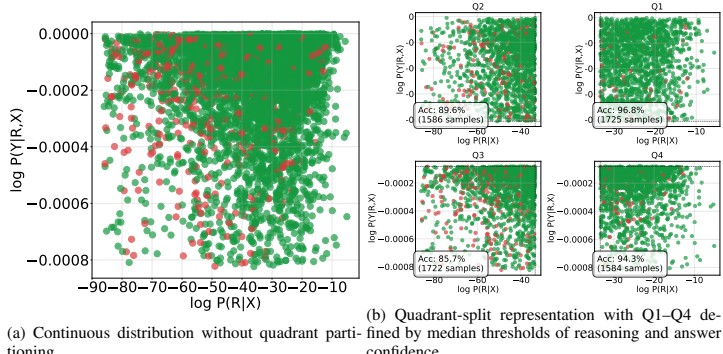

(a) Continuous distribution without quadrant partitioning.

(b) Quadrant-split representation with Q1–Q4 defined by median thresholds of reasoning and answer confidence.

Figure 11: Information Plane visualisations of Llama-3.1-8B on the GSM8K dataset ($k = 6$). Green indicates correct answers, Red indicates incorrect ones.

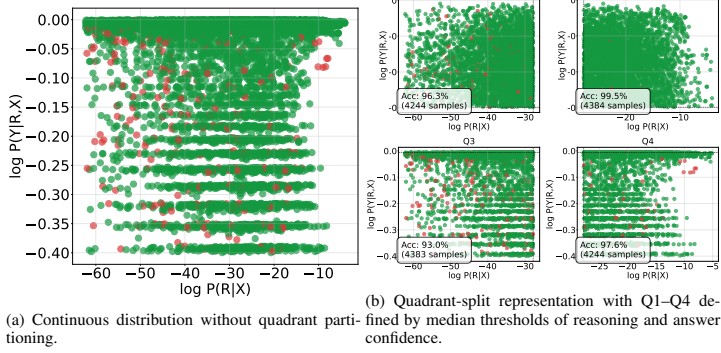

(a) Continuous distribution without quadrant partitioning.

(b) Quadrant-split representation with Q1–Q4 defined by median thresholds of reasoning and answer confidence.

Figure 12: Information Plane visualisations of Gemma-2-9B on the GSM8K dataset ($k = 6$). Green indicates correct answers, Red incorrect ones.

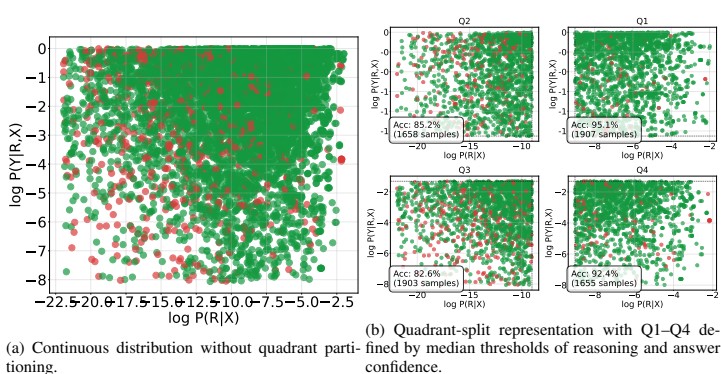

(a) Continuous distribution without quadrant partitioning.

(b) Quadrant-split representation with Q1–Q4 defined by median thresholds of reasoning and answer confidence.

Figure 13: Information Plane visualisations of Gemma-2-9B on the GSM8K dataset ($k = 6$). Green indicates correct answers, Red incorrect ones.

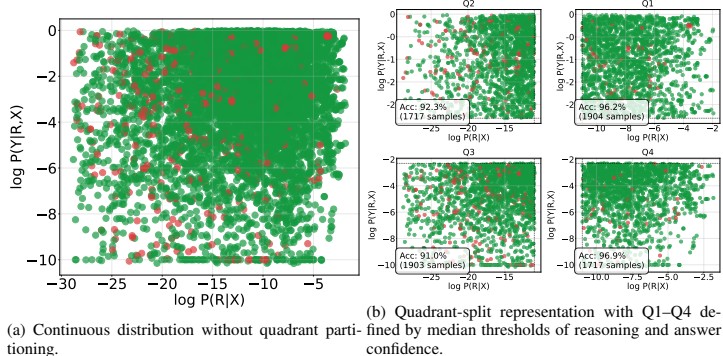

(a) Continuous distribution without quadrant partitioning.

(b) Quadrant-split representation with Q1–Q4 defined by median thresholds of reasoning and answer confidence.

Figure 14: Information Plane visualisations of Gemma-2-9B on the GSM8K dataset ($k = 6$). Green indicates correct answers, Red incorrect ones.

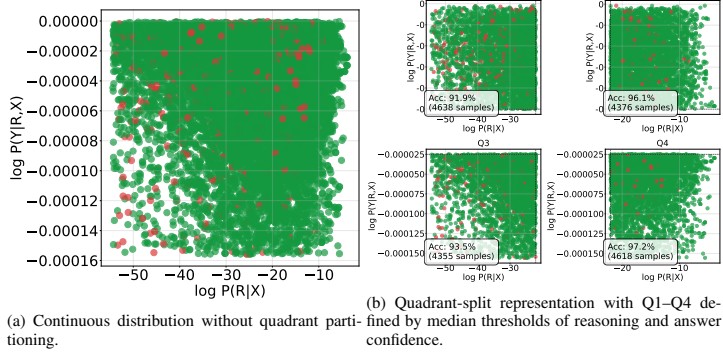

(a) Continuous distribution without quadrant partitioning.

(b) Quadrant-split representation with Q1–Q4 defined by median thresholds of reasoning and answer confidence.

Figure 15: Information Plane visualisations of Gemma-2-9B on the GSM8K dataset ($k = 6$). Green indicates correct answers, Red incorrect ones.

MATH500

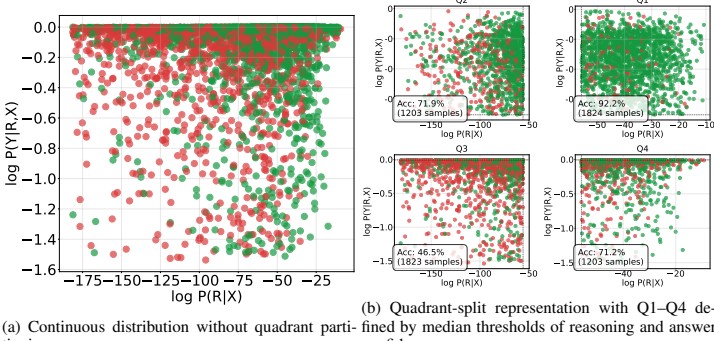

(a) Continuous distribution without quadrant partitioning.

(b) Quadrant-split representation with Q1–Q4 defined by median thresholds of reasoning and answer confidence.

Figure 16: Information Plane visualisations of Llama-3.1-8B on the MATH500 dataset ($k = 6$). Green indicates correct answers, Red incorrect ones.

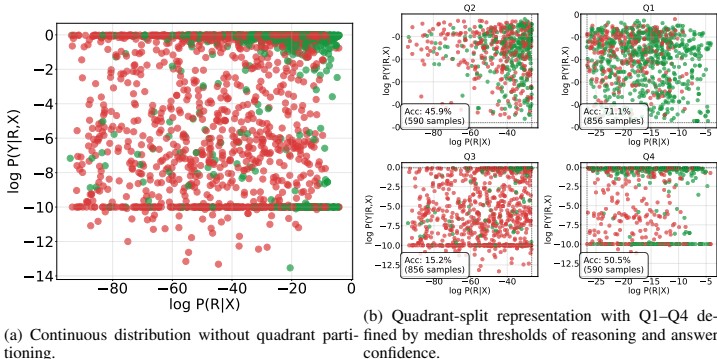

(a) Continuous distribution without quadrant partitioning.

(b) Quadrant-split representation with Q1–Q4 defined by median thresholds of reasoning and answer confidence.

Figure 17: Information Plane visualisations of Llama-3.1-70B on the MATH500 dataset ($k = 6$). Green indicates correct answers, Red incorrect ones.

(a) Continuous distribution without quadrant partitioning.

(b) Quadrant-split representation with Q1–Q4 defined by median thresholds of reasoning and answer confidence.

Figure 18: Information Plane visualisations of Gemma-2-9B on the MATH500 dataset ($k = 6$). Green indicates correct answers, Red indicates incorrect ones.

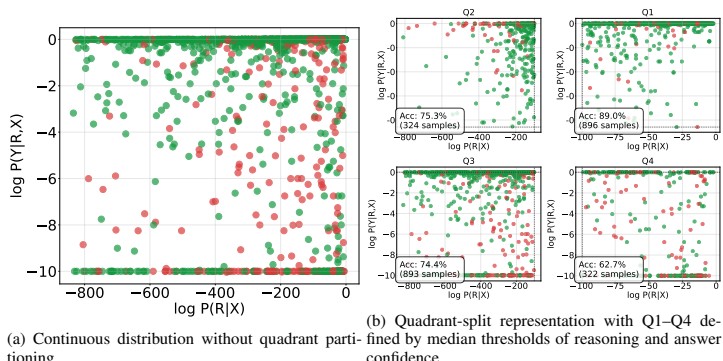

(a) Continuous distribution without quadrant partitioning.

(b) Quadrant-split representation with Q1–Q4 defined by median thresholds of reasoning and answer confidence.

Figure 19: Information Plane visualisations of Qwen3-8B on the MATH500 dataset ($k = 6$). Green indicates correct answers, Red incorrect ones.

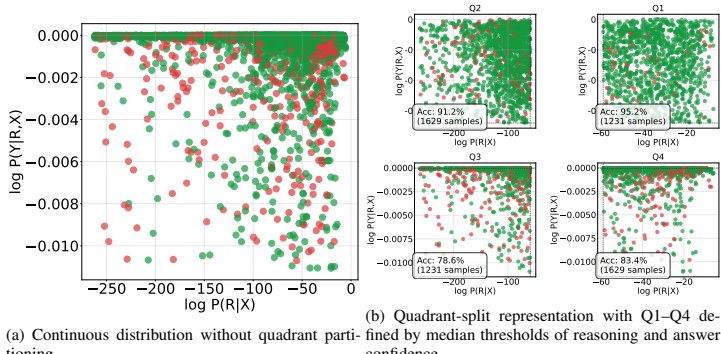

(a) Continuous distribution without quadrant partitioning.

(b) Quadrant-split representation with Q1–Q4 defined by median thresholds of reasoning and answer confidence.

Figure 20: Information Plane visualisations of Qwen3-8B on the MATH500 dataset ($k = 6$). Green indicates correct answers, Red incorrect ones.

AIME2024

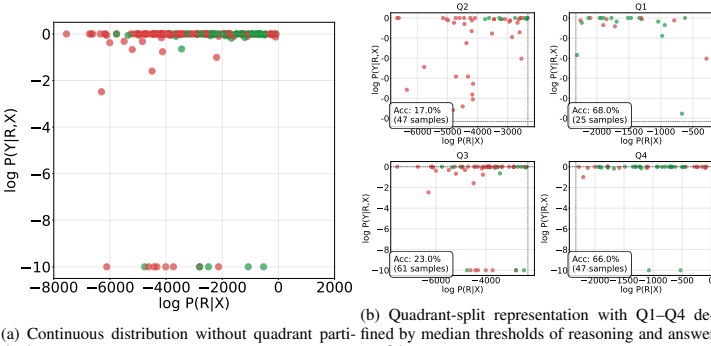

(a) Continuous distribution without quadrant partitioning.

(b) Quadrant-split representation with Q1–Q4 defined by median thresholds of reasoning and answer confidence.

Figure 21: Information Plane visualisations of DS-Distilled-Llama-8B on the AIME2024 dataset ($k = 6$). Green indicates correct answers, Red incorrect ones.

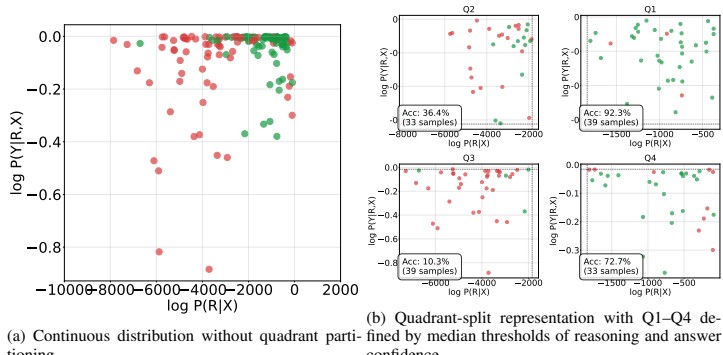

(a) Continuous distribution without quadrant partitioning.

(b) Quadrant-split representation with Q1–Q4 defined by median thresholds of reasoning and answer confidence.

Figure 22: Information Plane visualisations of DS-Distilled-Llama-8B on the AIME2024 dataset ($k = 6$). Green indicates correct answers, Red incorrect ones.

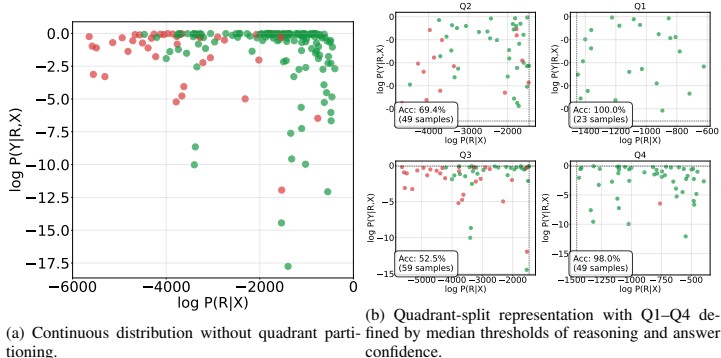

(a) Continuous distribution without quadrant partitioning.

(b) Quadrant-split representation with Q1–Q4 defined by median thresholds of reasoning and answer confidence.

Figure 23: Information Plane visualisations of DS-Distilled-Llama-8B on the AIME2024 dataset ($k = 6$). Green indicates correct answers, Red incorrect ones.

### E.1  75% THRESHOLD ON INFORMATION PLANE

As shown in Figure 24, increasing the confidence thresholds from the median to the 75th percentile isolates a region in the Information Plane with significantly higher accuracy, effectively identifying the most trustworthy solutions.

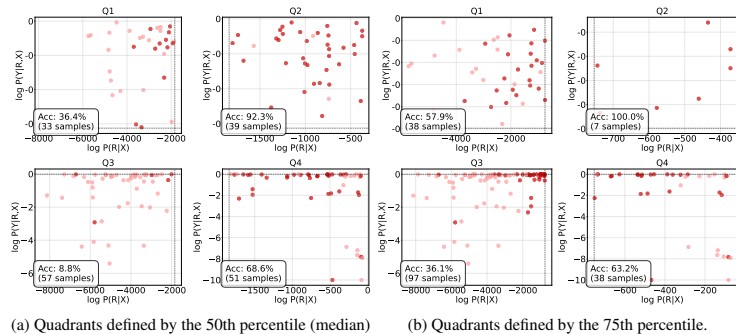

(a) Quadrants defined by the 50th percentile (median)

(b) Quadrants defined by the 75th percentile

Figure 24: Effect of confidence thresholding on the Information Plane for DS-Distilled-Qwen-2.5-7B ($k = 6$) on AIME2024.

## E.2 STATISTICAL TEST

| Term | t-statistic | p-value | U-statistic | Cohen's d | Mean (C/I) |
|---|---|---|---|---|---|
| $\log p(y \mid r, x)$ | 4.573 | 6.06e-6 | 38441.0 | 0.410 | -4.214 / -5.752 |
| $\log p(r \mid x)$ | 9.111 | 2.00e-18 | 45115.0 | 0.816 | -45.778 / -67.427 |

(a) Statistical tests for LLaMA-3.1-8B on Math500 dataset comparing correct (C) and incorrect (I) samples. All differences are highly significant ($p < 0.001$).

| Term | t-statistic | p-value | U-statistic | Cohen's d | Mean (C/I) |
|---|---|---|---|---|---|
| $\log p(y \mid r, x)$ | 5.759 | 1.48e-8 | 41596.0 | 0.539 | -0.411 / -1.470 |
| $\log p(r \mid x)$ | 6.992 | 8.76e-12 | 39096.0 | 0.655 | -39.870 / -53.686 |

(b) Statistical tests for LLaMA-3.1-70B on Math500 dataset comparing correct (C) and incorrect (I) samples. All differences are highly significant ($p < 0.001$).

| Term | t-statistic | p-value | U-statistic | Cohen's d | Mean (C/I) |
|---|---|---|---|---|---|
| $\log p(y \mid r, x)$ | 9.032 | 3.70e-18 | 42086.0 | 0.810 | -0.371 / -2.683 |
| $\log p(r \mid x)$ | 9.027 | 3.85e-18 | 45831.0 | 0.809 | -18.637 / -30.797 |

(c) Statistical tests for Gemma-2-9B on Math500 dataset comparing correct (C) and incorrect (I) samples. All differences are highly significant ($p < 0.001$).

| Term | t-statistic | p-value | U-statistic | Cohen's d | Mean (C/I) |
|---|---|---|---|---|---|
| $\log p(y \mid r, x)$ | 5.365 | 1.24e-7 | 36835.0 | 0.538 | -0.941 / -2.360 |
| $\log p(r \mid x)$ | 5.170 | 3.39e-7 | 31131.0 | 0.518 | -41.876 / -68.407 |

(d) Statistical tests for Qwen3-8B on Math500 dataset comparing correct (C) and incorrect (I) samples. All differences are highly significant ($p < 0.001$).

| Term | t-statistic | p-value | U-statistic | Cohen's d | Mean (C/I) |
|---|---|---|---|---|---|
| $\log p(y \mid r, x)$ | 6.090 | 2.26e-9 | 34499.5 | 0.640 | -0.378 / -1.816 |
| $\log p(r \mid x)$ | 4.979 | 8.81e-7 | 27660.0 | 0.523 | -61.918 / -95.847 |

(e) Statistical tests for Qwen3-32B on Math500 dataset comparing correct (C) and incorrect (I) samples. All differences are highly significant ($p < 0.001$).

| Term | t-statistic | p-value | U-statistic | Cohen's d | Mean (C/I) |
|---|---|---|---|---|---|
| $\log p(y \mid r, x)$ | 4.972 | 9.11e-7 | 27176.5 | 0.558 | -2.165 / -4.550 |
| $\log p(r \mid x)$ | 2.665 | 0.00795 | 21190.0 | 0.299 | -418.767 / -587.095 |

(f) Statistical tests for Think-Qwen3-8B on Math500 dataset (thinking enabled). Prediction and compression terms show significant differences between correct (C) and incorrect (I) samples.

| Term | t-statistic | p-value | U-statistic | Cohen's d | Mean (C/I) |
|---|---|---|---|---|---|
| $\log p(y \mid r, x)$ | 3.874 | 1.21e-4 | 29105.0 | 0.391 | -1.692 / -2.756 |
| $\log p(r \mid x)$ | 2.043 | 0.0416 | 29023.0 | 0.206 | -174.753 / -254.234 |

(g) Statistical tests for Think-DeepSeek-R1-Distill-Qwen-2.5-7B on Math500 dataset (thinking enabled). Prediction and compression terms show significant differences between correct (C) and incorrect (I) samples.

| Term | t-statistic | p-value | U-statistic | Cohen's d | Mean (C/I) |
|---|---|---|---|---|---|
| $\log p(y \mid r, x)$ | 5.991 | 4.00e-9 | 39822.0 | 0.565 | -0.973 / -3.196 |
| $\log p(r \mid x)$ | 4.634 | 4.60e-6 | 31908.0 | 0.437 | -246.181 / -500.004 |

(h) Statistical tests for Think-DeepSeek-R1-Distill-LLaMA-8B on Math500 dataset (thinking enabled). Prediction and compression terms show highly significant differences between correct (C) and incorrect (I) samples.

Table 12: Statistical tests on Math500 comparing correct (C) and incorrect (I) samples across multiple models. All show significant differences, though effect sizes vary.

## F   INTRA-MODEL RELIABILITY

To support the intra-model results in Section 5.2, we analyse the calibration of PiCSAR's confidence signal using the evaluation traces collected for the Qwen3 family. For every sample we pair the an-

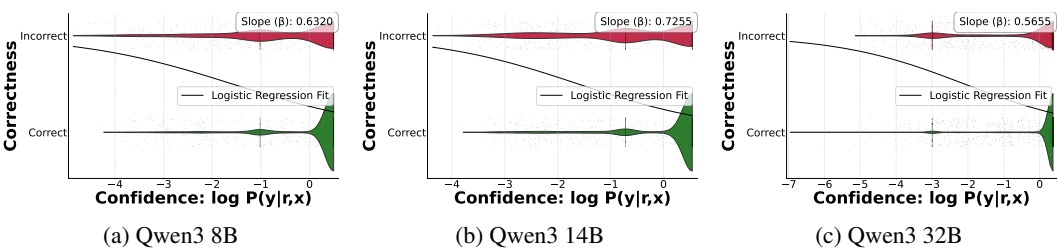

(a) Qwen3 8B  (b) Qwen3 14B  (c) Qwen3 32B

Figure 25: A detailed visualisation on the correct/incorrect densities based on logistic regression plot.

swer log-probability $\log p(y \mid r, x)$ with its correctness label and fit a separate model per backbone. The resulting calibration curves in Figure 5 exhibit a consistent monotonic trend: the logistic slopes are 0.63, 0.73, and 0.57 for Qwen3-8B, 14B, and 32B respectively, and the corresponding point-biserial coefficients ($r \approx 0.31, 0.35, 0.29$) show a positive correlation between higher confidence and the probability of a correct answer.

Figure 25 also shows how this effect manifests in the raw score distribution. Correct solutions concentrate around higher confidence values (closer to zero log-probability), whereas incorrect ones remain several nats lower, leaving limited overlap in the high-confidence region.

### F.1 LOGISTIC REGRESSION EXPERIMENTAL TRAINING

We model the relationship between confidence and correctness using logistic regression, similar to Gema et al. (2024). The binary outcome variable encodes whether the final answer is correct ($y \in 0, 1$), while the predictor is the model's confidence score expressed as the log-probability of the final answer:

$$Pr(y = 1 \mid Conf) = \sigma(\alpha + \beta \cdot Conf)$$

where $\sigma$ is the sigmoid function. The regression coefficient $\beta$ quantifies the change in log-odds of correctness per unit change in confidence. A positive $\beta$ indicates that higher confidence increases the likelihood of correctness. For instance, as shown in Figure 25b, in Qwen3-14B, $\beta = 0.7255$ corresponds to more than doubling the odds of correctness ($e^{0.7255} \approx 2.07$).

### F.2 POINT-BISERIAL CORRELATION COEFFICIENT

As a complementary measure to logistic regression, we compute the point-biserial correlation coefficient between confidence scores (continuous) and correctness (binary). This statistic, mathematically equivalent to Pearson's correlation with a dichotomous variable, directly quantifies the strength of association between the two. It is defined as

$$r = \frac{\bar{x}_1 - \bar{x}_0}{s_x} \sqrt{\frac{n_1 n_0}{n^2}},$$

where $\bar{x}_1$ and $\bar{x}_0$ denote the mean confidence scores for correct and incorrect samples, $s_x$ is the pooled standard deviation, and $n_1, n_0$ are the respective sample counts. The coefficient is bounded in $[-1, 1]$, with positive values indicating alignment between confidence and correctness. For instance, an $r$ of 0.35 for Qwen3-14B indicates a moderate positive association. Together with logistic regression, this provides a scale-free validation that confidence is a consistent predictor of correctness within a given model.

## G THE USE OF LARGE LANGUAGE MODELS (LLMS)

We have used LLM as a writing aid to assist with fluency and grammatical checking.

