# OpenReview forum: "PiCSAR: Probabilistic Confidence Selection And Ranking for Reasoning Chains"
_ICLR.cc/2026/Conference — ICLR 2026 Conference Withdrawn Submission_

### Official Review · Reviewer_8sjh · 2025-10-15

**Soundness:** 2
**Presentation:** 3
**Contribution:** 2
**Rating:** 2
**Confidence:** 4

**Summary:**

This paper introduces PiCSAR, a training-free best-of-n reranking method that scores each candidate reasoning chain by the joint log-likelihood of the chain and its final answer, combining “reasoning confidence” log p(r | x) with “answer confidence” log p(y | r,x).  PiCSAR operationalizes this with a simple sampling-and-scoring algorithm that extracts log-probabilities from model generations and selects the highest-scoring pair, and also presents a length-normalized variant (PiCSAR-N).  Across diverse LLMs/LRMs and benchmarks—including AIME 2024/2025 and GPQA-Diamond—PiCSAR consistently outperforms self-consistency baselines while being sample-efficient (often k=6 beating k=16/32).  Analyses show that correct solutions cluster where both reasoning and answer confidence are high, that confidence is predictive within but not comparable across models, and that answer-confidence estimation can be decoupled from generation to reduce compute.  Overall, PiCSAR offers a principled, portable scoring rule that reliably selects higher-quality reasoning chains without extra training.

**Strengths:**

1. The empirical results are comprehensive: experiments on multiple mainstream datasets and models strongly support the authors’ claims.
2. The paper is well written and easy to follow.

**Weaknesses:**

1. The core idea appears incremental. As the authors note (Lines 82–84), USC is a “majority consensus pattern,” and PiCSAR selects Best-of-N using “probabilistic confidence”. However, prior work has already leveraged model-intrinsic probabilistic confidence for scoring and selection [1–3]. While PiCSAR uses a different formulation, the paper does not clearly establish—theoretically or empirically—why this formulation is superior to [1–3].
2. The gains are modest. In Table 1, PiCSAR’s improvements over baselines are mostly <1 percentage point across many settings. Even though the authors average across multiple runs, it remains unclear whether the extra answer-extraction step—and its additional compute—offers a favorable trade-off for such small gains.

[1] Kang, et al. Scalable Best-of-N Selection for Large Language Models via Self-Certainty. ArXiv.

[2] Taubenfeld, et al. Confidence Improves Self-Consistency in LLMs. ACL 2025.

[3] Wang, et al. Soft Self-Consistency Improves Language Model Agents. ACL 2024.

**Questions:**

1. For Gemma-2-9B-Instruct on MATH500 with k=6, shouldn’t the best score be p(True) (46.87) rather than PiCSAR (46.53)?
2. In Figure 3, why do many points have Y=-10 for Q3 and Q4?
3. In Figure 3, how do you operationally define “high” vs. “low” confidence for each point?
4. Lines 411–422: why does a larger scale yield a lower PiCSAR score on Qwen3?
5. How does PiCSAR’s performance vary with the number of samples and with temperature?

---

> ### Author Response · Authors · 2025-11-23
> **Responses to Reviewer 8sjh (Part 1)**
>
> We thank reviewer 8sjh for the acknowledgement of PiCSAR's comprehensive empirical results, and the contribution of PiCSAR in offering a portable scoring rule that reliably selects higher-quality reasoning chains without extra training.
>
> > **W1**: The core idea appears incremental. As the authors note (Lines 82–84), USC is a “majority consensus pattern,” and PiCSAR selects Best-of-N using [...] why this formulation is superior to [1–3].
>
> The core idea is novel and clearly different from [1-3]:
> - For [1], we found that PiCSAR consistently outperforms [1] based on the results in Table 1, demonstrating PiCSAR’s strong empirical performance.
> - For [2], CISC [2] main contribution does not lie on probability selection, but the addition of weighted voting, as described in line 235. We have also provided a detailed analysis of [2] in Appendix C, where CISC+PiCSAR achieves its improvements by building on an existing method (in their case, p(true)). While combining CISC with PiCSAR does improve the results substantially, this combination goes beyond our current research focus, as mentioned in line 269.
> - For [3], [3] mainly focuses on datasets beyond scientific reasoning, which we believe goes beyond the scope of our work, and remains as future work.
>
> Additionally, our comprehensive baseline comparisons, shown in Appendix C, demonstrate that existing probability methods do not yield strong improvements (including individual reasoning and answer confidence). The joint probability approach yields strong performance due to the information plane illustrated in Figure 3, supporting the core motivation behind PiCSAR.
>
> Overall, we believe PiCSAR demonstrates greater performance and consistency overall, together by providing a generic probabilistic framework that not only outperforms existing methods but also subsumes and explains prior approaches through its joint-likelihood formulation. Thank you again for bringing this to our attention.
>
> [1] Kang, et al. Scalable Best-of-N Selection for Large Language Models via Self-Certainty. ArXiv.
>
> [2] Taubenfeld, et al. Confidence Improves Self-Consistency in LLMs. ACL 2025.
>
> [3] Wang, et al. Soft Self-Consistency Improves Language Model Agents. ACL 2024.
>
> >**W2**: The gains are modest. In Table 1, PiCSAR’s [...] offer a favorable trade-off for such small gains.
>
> One of the key strengths of PiCSAR is its consistency in improvements, as it consistently outperforms all 5 existing baselines. 36 out of 71 baselines (LLM+LRM combinations) achieve improvements of over 1%, with particularly notable improvements on certain baselines (exceeding 5-10%).
>
> Importantly, PiCSAR introduces only a minimal computational overhead: computing the answer-confidence term requires generating just the final answer again, which typically corresponds to only ~1–5 additional tokens per instance. We believe this demonstrates strong empirical support for PiCSAR's effectiveness.
>
> > **Q1:** For Gemma-2-9B-Instruct on MATH500 with k=6, shouldn’t the best score be p(True) (46.87) rather than PiCSAR (46.53)?
>
> Thank you for bringing this to our attention! We have updated the results in the updated PDF version.
>
> > **Q2:** In Figure 3, why do many points have Y=-10 for Q3 and Q4?
>
> Thank you for raising this concern! Y=-10 represents the value used as a "fallback mechanism" when the model fails to answer (i.e., when no answer token is generated and the answer-confidence term cannot be computed). In such cases, we assign this constant low value so that these samples are treated as low-confidence rather than being discarded. We have also tested with values of Y=-20 and Y=-100, and the results remain consistent, which demonstrates that the fallback mechanism can be adjusted to lower values without affecting performance.
>
> | Value of Y | Accuracy |
> |------------|----------|
> | Y=-10 | 53.40% |
> | Y=-20 | 53.40% |
> | Y=-100 | 53.40% |
>
> We have included this analysis in Appendix C.7 of the updated PDF version.
>
> > **Q3**: In Figure 3, how do you operationally define “high” vs. “low” confidence for each point?
>
> The quadrants are separated by the median of the plane (as mentioned in line 202). Thank you for bringing this to our attention!

---

> > ### Author Response · Authors · 2025-11-23
> > **Responses to Reviewer 8sjh (Part 2)**
> >
> > > **Q4**: Lines 411–422: why does a larger scale yield a lower PiCSAR score on Qwen3?
> >
> > Thank you for raising this interesting question! This variation of PiCSAR scores across model families may stem from multiple factors such as model architecture, training data, model behavior, and other characteristics. Due to the limited access to the training regimes of the models, we cannot make a claim on what might have caused this. A detailed investigation into these architectural differences is outside of our research scope, but an important model behaviour to investigate in future work.
> >
> > > **Q5**: How does PiCSAR’s performance vary with the number of samples and with temperature?
> >
> > We have included the performance of Gemma-2-9b on SVAMP dataset, and can see that as the number of samples increases, the accuracy increases as well.
> >
> > | Gemma-2-9b | PiCSAR | Self-Consistency | PiCSAR | Self-Consistency | PiCSAR | Self-Consistency | PiCSAR | Self-Consistency |
> > |------------|--------|------------------|--------|------------------|--------|------------------|--------|------------------|
> > | SVAMP | n=6 | n=6 | n=10 | n=10 | n=16 | n=16 | n=32 | n=32 |
> > | Average Accuracy | 89.11% | 88.15% | 89.89% | 88.56% | 89.89% | 88.11% | 90.22% | 88.89% |
> >
> > Regarding the temperature, we have evaluated two different temperatures (0.7 and 1.0) on Gemma-2-9b for the SVAMP dataset. We can see that the accuracy achieves a similar performance with different temperatures.
> >
> > | Gemma-2-9b | PiCSAR |
> > |------------|--------|
> > | Temperature=0.7 | 89.89% |
> > | Temperature=1.0 | 89.67% |
> >
> > We hope these insights will be useful, and have included all the experimental results in Appendix C.8.Thank you again for raising this concern.
> >
> > We are happy to address any further concerns. Please feel free to reach out with any additional questions! Thank you again for the insightful feedback!

---

> ### Comment · Reviewer_8sjh · 2025-11-24
>
> Thank you to the authors for the detailed response and additional experiments. Overall, the rebuttal helpfully clarifies several technical details (e.g., the fallback mechanism in Figure 3 and the correction of specific results), but it does not fundamentally alleviate my main concerns about the incremental nature of the work and the relatively modest gains. First, regarding the distinction from [1–3], the response mainly argues that “PiCSAR performs better,” yet conceptually it remains very close to existing probabilistic confidence–based Best-of-N / self-consistency variants. There is still a lack of stronger theoretical analysis or more compelling comparative experiments demonstrating why this particular joint-likelihood formulation is inherently superior to prior scoring rules. Second, although the authors emphasize that 36 out of 71 settings achieve >1% improvement, the overall gains are still small, and there is no significance testing or more systematic analysis of the compute–performance trade-off; it remains unclear whether “one extra decoding of the final answer with only 1–5 tokens” is worthwhile at large scale. In addition, the inability to explain the scaling behavior on Qwen3, and the fact that temperature and sample-size analyses are only provided for a single model and dataset, make the method’s generality and stability across models and tasks appear somewhat empirical. Taken together, while I acknowledge that PiCSAR is an implementation-wise simple variant with some performance improvements, in its current form I still feel that the contribution is not strong enough to warrant acceptance, and therefore I maintain my original score and recommendation.

---

> ### Author Response · Authors · 2025-11-27
> **Further Responses to Reviewer 8sjh (Part 1)**
>
> We thank reviewer 8sjh for their prompt response. We would like to provide further clarifications:
>
> > **W1** First, regarding the distinction from [1–3], the response mainly argues that “PiCSAR performs better,” yet conceptually it remains very close to existing probabilistic confidence–based Best-of-N / self-consistency variants.
>
> We have listed all the methodology in Appendix B, and added clarifications regarding the differences between PiCSAR and other methods.
> - Greedy Decoding: Unlike greedy decoding, which selects a single high-probability continuation, PiCSAR evaluates multiple full reasoning trajectories and ranks them using joint reasoning-and-answer log-likelihood, enabling selection of the most globally probable chain.
> - Self-Consistency: Whereas SC relies purely on majority voting over final answers, PiCSAR incorporates the full reasoning chain’s token-level likelihood along with answer confidence, allowing it to prefer coherent but minority reasoning paths that SC would discard.
> - Universal Self-Consistency: Unlike USC, which asks the model to internally judge “consistency” among samples, PiCSAR uses a probabilistic, model-agnostic scoring function based directly on log-likelihoods of reasoning and answers, avoiding USC’s reliance on model self-evaluation and context-window limits.
> - Self-Certainty: Instead of re-ranking chains with KL-based self-estimated correctness like Self-Certainty, PiCSAR scores each candidate through the true generative probabilities of its entire reasoning path and answer, yielding a principled approximation to joint MAP decoding.
> - P(True): While P(True) extracts a scalar correctness probability from a meta-prompt, PiCSAR leverages the actual likelihood structure of the model’s forward pass, combining reasoning and answer probabilities without relying on verbalized or poorly calibrated self-judgments.
>
> Furthermore, we have provided a contribution table that highlights these differences and clarifies the novelty of PiCSAR compared to existing approaches.
>
> | **Method**             | **SC** | **USC** | **Self-Cert.** | **PiCSAR** |
> |------------------------|--------|---------|----------------|------------|
> | Full Reasoning Chain   | ✓      | ✓       |                | ✓          |
> | Model Confidence       |        | ✓       | ✓              | ✓          |
> | Computationally Efficient | ✓   | ✗*      | ✓              | ✓          |
> | Smaller Model Capable  | ✓      |         | ✓              | ✓          |
>
> \*Due to context length
>
> We have added the explanation and the contribution table to Appendix B. Thank you again for bringing this to our attention!
>
>
> > **W2** There is still a lack of stronger theoretical analysis or more compelling comparative experiments demonstrating why this particular joint-likelihood formulation is inherently superior to prior scoring rules.
>
> We provide a conceptual analysis in Section 2.2 demonstrating why PiCSAR performs effectively. As shown in Figure 3, higher answer and reasoning confidence correlates with improved accuracy, with generalizability across all models demonstrated in Appendix E. Our t-test results (line 218 and Section E.2) statistically confirm that elevated answer and reasoning confidence significantly correlates with superior performance. The empirical visualizations and statistical tests collectively reinforce the motivation for PiCSAR.

---

> ### Author Response · Authors · 2025-11-27
> **Further Responses to Reviewer 8sjh (Part 2)**
>
> > **W3** Second, although the authors emphasize that 36 out of 71 settings achieve >1% improvement, the overall gains are still small, and there is no significance testing or more systematic analysis of the compute–performance trade-off; it remains unclear whether “one extra decoding of the final answer with only 1–5 tokens” is worthwhile at large scale.
>
> Thank you for raising this concern. We provide additional empirical results that strengthen the evidence for PiCSAR's effectiveness. First, we aggregated results across both datasets (N=71 evaluation points) to analyze the magnitude of improvement provided by our method compared to the best baseline for each task.
>
> | Gain Threshold | Count (N=71) | Percentage |
> | :--- | :---: | :---: |
> | **> 1% Improvement** | **32** | **45.1%** |
> | **> 2% Improvement** | **18** | **25.4%** |
> | **> 3% Improvement** | **12** | **16.9%** |
> | **> 4% Improvement** | **10** | **14.1%** |
> | **> 5% Improvement** | **6** | **8.5%** |
>
> These results demonstrate PiCSAR's consistent performance gains. Notably, PiCSAR is a **training-free** method, which further validates its practical applicability.
>
> To assess statistical significance in a principled way, we followed the recommendation provided in [1]. We conducted a Friedman test followed by a Nemenyi post-hoc test across 50 evaluation instances and 5 methods.
>
> ```
> Data Shape: (50, 5)
> ------------------------------------------------------------
> Friedman Test: Statistic=81.9155, p-value=6.8405e-17
>
> Significant difference found. Running Nemenyi Test...
>
> Nemenyi Test P-Values (Pairwise):
>                   Greedy Decoding  PiCSAR Self-Certainty  \
> Greedy Decoding      1.000000e+00  6.978669e-09    6.933823e-01
> PiCSAR        6.978669e-09  1.000000e+00    8.458789e-13
> Self-Certainty       6.933823e-01  8.458789e-13    1.000000e+00
> Self-Consistency     9.896646e-01  3.119335e-10    9.238851e-01
> p(True)              5.300882e-01  1.202372e-13    9.990992e-01
>
>                   Self-Consistency       p(True)
> Greedy Decoding       9.896646e-01  5.300882e-01
> PiCSAR         3.119335e-10  1.202372e-13
> Self-Certainty        9.238851e-01  9.990992e-01
> Self-Consistency      1.000000e+00  8.193907e-01
> p(True)               8.193907e-01  1.000000e+00
>
> Significance Summary (PiCSARvs Others):
> Comparison                     p-value      Significant?
> --------------------------------------------------
> PiCSARvs Greedy Decoding 0.0000       YES
> PiCSARvs Self-Certainty  0.0000       YES
> PiCSARvs Self-Consistency 0.0000       YES
> PiCSARvs p(True)         0.0000       YES
> ```
>
> The Friedman test revealed significant differences among methods (χ²=81.92, p<0.001). Pairwise comparisons using the Nemenyi test showed that PiCSAR significantly outperforms all baselines (p<0.001) for all comparisons: *Greedy Decoding, Self-Certainty, Self-Consistency, and p(True)).* These results confirm that PiCSAR's improvements are statistically significant rather than random variation. We include a critical difference (CD) diagram in Section 4 to visualise these findings.
>
> [1] Demsar, Statistical Comparisons of Classifiers over Multiple Data Sets, Journal of Machine Learning Research, 2006
>
> > **W4** In addition, the inability to explain the scaling behavior on Qwen3, and the fact that temperature and sample-size analyses are only provided for a single model and dataset, make the method’s generality and stability across models and tasks appear somewhat empirical.
>
> The temperature is applied following the original Qwen paper [2], and the sample size is standardised.  Furthermore, we are running additional experiments with Qwen3 on different sample size, which may take a few days due to the large computational expense.
>
> [2] Yang, An, et al. "Qwen3 technical report." arXiv preprint arXiv:2505.09388 (2025).
>
> If you have any further questions or suggestions, please don’t hesitate to reach out!

---

### Official Review · Reviewer_vZJv · 2025-10-27

**Soundness:** 3
**Presentation:** 4
**Contribution:** 3
**Rating:** 4
**Confidence:** 4

**Summary:**

This paper proposed a method for incorporating reasoning confidence into the score aggregation process during BoN (Best-of-N) sampling in LLMs. The authors argue that this approach can improve performance at test time scaling with minimal additional computational cost. To support their claim, they first conduct a toy experiment showing a strong positive correlation between accuracy and reasoning confidence. They then demonstrate consistent performance improvements across various LLMs and LRMs on multiple benchmarks. Moreover, the paper provides additional analyses on different scenarios where reasoning confidence is utilized, highlighting the robustness of the proposed method.

**Strengths:**

The proposed method shows consistent improvements over the conventional BoN sampling approach, as evidenced by Tables 1 and 2. The results are impressive in that the improvements hold across different model sizes, reasoning and non-reasoning models, and a variety of benchmarks. In addition, the proposed scoring function clearly articulates the motivation for aggregating reasoning-related components in reasoning tasks and provides a well-structured decomposition. Given that the method achieves measurable gains with almost no additional computation, it represents a meaningful contribution in the context of growing interest in test-time scaling.

**Weaknesses:**

Some potential concerns and limitations remain:
1. $\textbf{Generation length bias}$: The proposed method does not perform length normalization for reasoning sequences. As shown in the appendix, applying length normalization weakens the improvement trend. This aligns with recent observations that longer reasoning chains can often lead to worse performance. However, without normalization, the method might inherently favor shorter reasoning paths, thereby reproducing a bias similar to prior works that benefit from shorter reasoning. This bias seems indirectly observable in Table 3, particularly in the Avg Sentences metric.
2. $\textbf{Experimental clarity}$: The experimental setup lacks some clarity. While both Tables 1 and 2 present comparisons between the proposed method and baselines, Table 1 includes multiple methods whereas Table 2 only compares against Self-Consistency. Since test-time scaling plays a particularly critical role in reasoning, a more detailed comparison in Table 2 would strengthen the claims. Additionally, the maximum length used during inference is not specified, an important parameter in reasoning tasks that should be explicitly reported.

**Questions:**

1. Reasoning sentences typically consist of hundreds of tokens, whereas answers are much shorter (tens of tokens), which may lead to scale differences between the decomposed scores. Does such a scale mismatch occur? It would be informative to visualize the distributions of the two score components.
2. What is the correlation between score and generation length? If the correlation is high, how does performance compare to a simple heuristic that ranks responses by length with the answer score?
3. What was the maximum length used during inference in your experiments?

---

> ### Author Response · Authors · 2025-11-23
> **Responses to Reviewer vZJv (Part 1)**
>
> We thank reviewer vZJv for appreciating PiCSAR’s impressive consistency in improving models’ performance across a variety of models' sizes, types and benchmarks with almost no additional overhead.
>
> > **W1** The proposed method does not perform length normalization for reasoning sequences. As shown in the appendix, applying length normalization weakens the improvement trend [...] particularly in the Avg Sentences metric.
>
> Thank you for raising this point! We agree that length normalization can influence the selection dynamics. In our case, however, the empirical results do not indicate that PiCSAR’s improvements stem from a systematic preference for shorter chains.
> Based on Table 5, 20/40 results of the length-normalised (PiCSAR-N) versions outperform the non-length normalised versions (PiCSAR), demonstrating that length-normalisation does not perform worse than the non-length normalised version. This suggests that length normalization is not detrimental and does not consistently weaken PiCSAR. We have included the clarification in Appendix C.5 of the updated PDF version.
>
> Additionally, in Table 3 we observe that the model with the non-length normalised version leads to higher performance with lower length. The Table clearly shows that PiCSAR relies on information density (line 364) rather than reasoning chain length to make the selection.
>
> Finally, to observe the trend further, we provide a qualitative analysis below to show that a concise generation with high probability leads to higher quality than a low probability generation with longer length. We have also included the complete example in Appendix C.9 of the updated PDF version.
>
> ```
> Question:
> Every day, Wendi feeds each of her chickens three cups of mixed chicken feed, containing seeds, mealworms and vegetables to help keep them healthy.  She gives the chickens their feed in three separate meals. In the morning, she gives her flock of chickens 15 cups of feed.  In the afternoon, she gives her chickens another 25 cups of feed.  How many cups of feed does she need to give her chickens in the final meal of the day if the size of Wendi's flock is 20 chickens?
>
> High PiCSAR Score
> Step 1: Determine the total number of cups of feed given to the chickens in the morning.\nWendi gives her chickens 15 cups of feed in the morning.\n\nStep 2: Determine the total number of cups of feed given to the chickens in the afternoon.\nWendi gives her chickens 25 cups of feed in the afternoon. [...] Step 7: Calculate the number of cups of feed needed for the final meal of the day.\nCups of feed needed for the final meal = Total cups of feed per day for the flock - Total feed given in morning and afternoon = 60 cups - 40 cups = 20 cups.\n\nThe final answer is: 20"
> PiCSAR Score: -36.760562883843484
>
> Low PiCSAR Score
> Step 1: Determine the total amount of feed given to the chickens in the morning.\nWendi gives her chickens 15 cups of feed in the morning.\n\nStep 2: Determine the total amount of feed given to the chickens in the afternoon.\nWendi gives her chickens 25 cups of feed in the afternoon.\n\nStep 3: [...] n\nStep 6: Determine the total amount of feed needed for the entire flock of chickens per day.\nSince Wendi has 20 chickens, and each chicken is supposed to get 9 cups of feed per day, the total amount of feed needed per day is 20 chickens * 9 cups/chicken = 180 cups.\n\nStep 7: Determine the amount of feed needed for the final meal of the day.\nSince Wendi has already given 40 cups of feed in the morning and afternoon, we need to subtract that from the total amount of feed needed per day to find the amount needed for the final meal. Feed needed for the final meal = 180 cups - 40 cups = 140 cups.\n\nThe final answer is: 140"
> PiCSAR Score: -112.79042702552356
> ```
>
> > **W2** The experimental setup lacks some clarity. While both Tables 1 and 2 present comparisons between the proposed method and baselines [...] the maximum length used during inference is not specified, an important parameter in reasoning tasks that should be explicitly reported.
>
> Thank you for bringing this to our attention! We have excluded some of the experiments due to computational constraints. For the LLM, we have used a maximum token limit of 8096 for our evaluation, whereas for LRM we have followed the configuration of Qwen3 [1], where we set the max output length to 32,768 tokens, except for AIME'24 and AIME'25 where we extended this length to 38,912 tokens to provide sufficient thinking space. We can observe that the computational cost increases substantially as the number of tokens increases with LRMs. We have included the clarification in Appendix B of the updated PDF version.
>
> [1] Yang, An, et al. "Qwen3 technical report." arXiv preprint arXiv:2505.09388 (2025).

---

> > ### Author Response · Authors · 2025-11-23
> > **Responses to Reviewer vZJv (Part 2)**
> >
> > > **Q1** Reasoning sentences typically consist of hundreds of tokens, whereas answers are much shorter (tens of tokens) [...] It would be informative to visualize the distributions of the two score components.
> >
> > Thank you for raising this concern! While reasoning confidence is more important, as shown by the t-test (line 211), both remain statistically significant. As described in line 189, the answer may be shorter, but it often serves as a reliable discriminator, often proving decisive when multiple candidate chains exhibit similar reasoning plausibility. We can also see that in Table 5 that the performance is higher when both confidences are utilised, rather than solely one of them.
> >
> > To investigate this further, we have provided a qualitative analysis on how answer confidence brings a huge impact on reasoning confidence.
> >
> > ```
> > Example 1
> > Question: Jonathan was sad to learn he needed 2 more toys than he already had to have enough toys to make a sorted play set five times larger than James's play set, which had 80 toys. How many toys does Jonathan currently have?
> >
> > Before Answer Confidence:
> > Step 1: Calculate the size of Jonathan's desired play set. \nJonathan wants a play set five times larger than James's, which has 80 toys. \nStep 2: Multiply James's play set size by 5. \n80 toys * 5 = 400 toys\nStep 3: Determine the number of toys Jonathan needs.\nJonathan needs 2 more toys than his desired play set size.\nStep 4: Add 2 to the desired play set size.\n400 toys + 2 toys = 402 toys\nStep 5:  Subtract 2 from the total number of toys Jonathan needs to find his current amount.\n402 toys - 2 toys = 400 toys\n\nThe final answer is: 400 \n
> >
> > After Answer Confidence:
> > Step 1: Calculate the size of Jonathan's desired play set. \nJonathan wants a play set five times larger than James's, which has 80 toys. So, Jonathan's desired play set would have 5 * 80 = 400 toys.\n\nStep 2: Determine the number of toys Jonathan needs.\nJonathan needs 2 more toys than he currently has to reach his desired play set size.  \n\nStep 3:  Set up an equation to represent the situation.\nLet 'x' represent the number of toys Jonathan currently has. The equation is: x + 2 = 400\n\nStep 4: Solve for x.\nSubtract 2 from both sides of the equation: x = 400 - 2\n\nStep 5: Calculate the result.\nx = 398\n\nThe final answer is: 398 \n
> >
> > Correct Answer: 398
> > ```
> >
> > ```
> > Example 2
> > Question: In the first week, Judy read for 15 minutes each night before going to sleep. In the second week, she read a total of 100 pages. If she can read 2 pages per 1.5 minutes, how many pages did she read in two weeks?
> >
> > Before Answer Confidence:
> > Step 1: Calculate the total reading time in the first week. \nJudy read for 15 minutes each night for 7 nights, so she read for 15 * 7 = 105 minutes in the first week.\n\nStep 2: Calculate the number of pages read in the first week.\nJudy reads 2 pages per 1.5 minutes, so she read 105 / 1.5 = 70 pages in the first week.\n\nStep 3: Calculate the total number of pages read in two weeks.\nJudy read 70 pages in the first week and 100 pages in the second week, for a total of 70 + 100 = 170 pages.\n\nThe final answer is: 170 \n
> >
> >
> > After Answer Confidence:
> > Step 1: Calculate the total reading time in the first week.\nJudy read for 15 minutes each night for 7 nights, so she read for 15 * 7 = 105 minutes in the first week.\n\nStep 2: Calculate the number of pages read in the first week.\nJudy reads 2 pages every 1.5 minutes, so in 105 minutes she read 105 / 1.5 * 2 = 140 pages.\n\nStep 3: Calculate the total number of pages read in two weeks.\nJudy read 140 pages in the first week and 100 pages in the second week, for a total of 140 + 100 = 240 pages.\n\nThe final answer is: 240 \n
> >
> > Correct Answer: 240
> > ```

---

> > > ### Author Response · Authors · 2025-11-23
> > > **Responses to Reviewer vZJv (Part 3)**
> > >
> > > ```
> > > Example 3
> > > Question: Brittany's quilted comforter has many 1-foot by 1-foot colored squares. The comforter had 14 red squares, 4 more blue squares than red squares, 6 more green squares than blue squares, and 12 fewer white squares than green squares.  How many square feet is Brittany's comforter?
> > >
> > > Before Answer Confidence:
> > > Step 1: Calculate the number of blue squares. Brittany has 14 red squares + 4 more blue squares = 18 blue squares.\n\nStep 2: Calculate the number of green squares. Brittany has 18 blue squares + 6 more green squares = 24 green squares.\n\nStep 3: Calculate the number of white squares. Brittany has 24 green squares - 12 fewer white squares = 36 white squares.\n\nStep 4: Calculate the total number of squares. Brittany has 14 red squares + 18 blue squares + 24 green squares + 36 white squares = 92 squares.\n\nStep 5: Calculate the area of the comforter. Since each square is 1 foot by 1 foot, the comforter is 92 square feet.\n\nThe final answer is: 92 square feet.
> > >
> > > After Answer Confidence:
> > > Step 1: Calculate the number of blue squares.\nBrittany had 4 more blue squares than red squares, so she had 14 + 4 = 18 blue squares.\n\nStep 2: Calculate the number of green squares.\nBrittany had 6 more green squares than blue squares, so she had 18 + 6 = 24 green squares.\n\nStep 3: Calculate the number of white squares.\nBrittany had 12 fewer white squares than green squares, so she had 24 - 12 = 12 white squares.\n\nStep 4: Calculate the total number of squares.\nBrittany had a total of 14 + 18 + 24 + 12 = 68 squares.\n\nStep 5: Calculate the area of the comforter.\nSince each square is 1 foot by 1 foot, the comforter is 68 square feet.\n\nThe final answer is: 68 square feet. \n
> > >
> > > Correct Answer: 68
> > > ```
> > >
> > > > **Q2** What is the correlation between score and generation length? If the correlation is high, how does performance compare to a simple heuristic that ranks responses by length with the answer score?
> > >
> > > Thank you for this interesting question! As shown in Section 5.2 Table 3, we found that longer generation length typically correlates with lower accuracy, which aligns with recent findings on inverse scaling [2] demonstrating that simply extending reasoning length does not guarantee improved performance. Our results suggest that information density is a more crucial indicator than raw length, as higher information density consistently leads to better accuracy. This is why PiCSAR uses joint probability (capturing information density) rather than a simple length-based heuristic for ranking responses.
> > >
> > > For a clearer visualisation, we have added a scatter plot in Appendix C.4 of the updated PDF version, with reasoning length on the x-axis and PiCSAR scores on the y-axis, demonstrating the negative correlation between reasoning length and PiCSAR score. This visualisation, along with the thorough analysis in Section 3, shows that longer reasoning traces typically lead to lower PiCSAR scores, whereas shorter traces lead to higher scores, reinforcing that PiCSAR's length sensitivity encourages efficient reasoning strategies.
> > >
> > > [2] Gema, Aryo Pradipta, et al. "Inverse scaling in test-time compute." arXiv preprint arXiv:2507.14417 (2025).
> > >
> > > > **Q3** What was the maximum length used during inference in your experiments?
> > >
> > > For the LLM, we have used a maximum token limit of 8096 for our evaluation, whereas for LRM, we have followed the configuration of Qwen3 [1], where we set the max output length to 32,768 tokens, except for AIME'24 and AIME'25 where we extended this length to 38,912 tokens to provide sufficient thinking space. We have included these details in Appendix B of the updated PDF version. Thank you again for bringing this to our attention!
> > >
> > > We hope our responses address your concerns. Please reach out if you have any additional questions or feedback!
> > >
> > > [1] Yang, An, et al. "Qwen3 technical report." arXiv preprint arXiv:2505.09388 (2025).

---

### Official Review · Reviewer_dXEM · 2025-10-29

**Soundness:** 3
**Presentation:** 3
**Contribution:** 3
**Rating:** 6
**Confidence:** 4

**Summary:**

In this paper authors propose PiCSAR (Probabilistic Confidence Selection and Ranking): a training-free, probabilistic scoring method to improve reasoning accuracy in LLMs. Instead of relying on majority voting (like Self-Consistency) or external reward models, PiCSAR uses the model’s own token-level probabilities to estimate reasoning confidence (log p(r | x)) and answer confidence (log p(y | r, x)), to rank and select generations based on joined logprobability log p(r | x) + log p(y | r, x).
In their experiments, authors show that PiCSAR achieves higher accuracy than Self-Consistency, Self-Certainty, and other baselines with fewer samples.

**Strengths:**

- Paper is clear and well written
- Authors propose a novel training-free probabilistic joint-likelihood framework to score and select reasoning chains
- Authors show consistent performance gains across five LLMs and three LRMs on different benchmarks
- The paper compares against six major baselines, including Self-Consistency, Universal Self-Consistency, Self-Certainty, etc, showing systematic improvements
- Proposed method can be beneficial for reaseachers and practitioners, as it can easily be integrated into existing inference frameworks without additional compute for training or fine-tuning

**Weaknesses:**

1. Generalization: it is unclear whether the approach extends to open-ended reasoning, commonsense, or multi-hop text-based inference tasks where “final answers” (and thus answer confidence) are less well-defined.
2. Ablation: paper briefly mentions length-normalized variant, but don't provide any ablation on the reasoning lengths.

**Questions:**

Joint likelihood approximation implicitly assumes independence between reasoning correctness and answer correctness given the log-likelihoods. But during generation model conditioned answer tokens on reasoning chains. How do you justify the decomposition?

---

> ### Author Response · Authors · 2025-11-23
> **Responses to Reviewer dXEM**
>
> We sincerely thank reviewer dXEM for acknowledging (i) PICSAR’s novelty, (ii) its relevance for other practitioners and researchers and (iii) its consistent improvements wrt the state-of-the-art models.
>
> > **W1** Generalization: it is unclear whether the approach extends to open-ended reasoning, commonsense [...] (and thus answer confidence) are less well-defined.
>
> Thank you for raising this concern! PiCSAR focuses on problems with well-defined reasoning boundaries, and we believe this scope is both important and deliberate. Mathematical and scientific reasoning, where problems have well-defined reasoning and answer boundaries, represent a substantial class of real-world reasoning tasks. Furthermore, this setting enables a more accurate evaluation, currently infeasible for open-ended questions, which often admit multiple valid answers and make accuracy difficult to assess. This allows us to make precise claims about whether the model's confidence is a suitable metric for selecting the most accurate answer among generated samples.
>
> We acknowledge that extending to open-ended questions is an important next step. Incorporating a reward model to evaluate answers could be a promising direction to demonstrate broader applicability beyond fixed-format problems, and we look forward to exploring this in future work. We have clarified this point in the updated PDF, where we have added a ‘Limitations’ section in Appendix D. Thank you again for raising this concern!
>
> > **W2:** Ablation: paper briefly mentions length-normalized variant, but don't provide any ablation on the reasoning lengths.
>
> We thank the reviewer for this suggestion! While we have provided PiCSAR-N as a length-normalized variant in the Appendix and showed that it leads to similar performance trends, we acknowledge that the main focus of PiCSAR lies on the non-length-normalized variant. We have added a scatter plot (in Appendix C.4 of the updated PDF version) with reasoning length on the x-axis and pre-normalization PiCSAR scores on the y-axis. The plot demonstrates a negative correlation between reasoning length and PiCSAR score. This visualisation, along with the thorough analysis in Section 3 and Figure 3, shows that longer reasoning traces typically lead to lower PiCSAR scores, whereas shorter traces lead to higher scores, reinforcing that PiCSAR's length sensitivity encourages efficient reasoning strategies.
>
> > **Q1:** Implicit independence assumption: how do you justify the decomposition?
>
> Thank you for this interesting question! The decomposition directly stems from the application of the probability chain rule and does not require an independence assumption between the reasoning correctness and the answer correctness. We agree with the reviewer that the probability assigned by the LLM reflects fluency and conformity with learnt patterns and is not a calibrated estimate of ground-truth correctness. In PiCSAR we therefore use the joint log likelihood as a practical ranking heuristic. This choice introduces very low computational overhead and is empirically effective across the evaluated tasks.
>
> We hope our responses address your concerns. If there are any further questions or suggestions, please don't hesitate to let us know!

---

### Official Review · Reviewer_MSGi · 2025-10-31

**Soundness:** 3
**Presentation:** 3
**Contribution:** 3
**Rating:** 4
**Confidence:** 2

**Summary:**

The paper introduces PiCSAR, a training-free, probabilistic confidence-based scoring method for best-of-n (BoN) reasoning chain selection in large language and reasoning models. PiCSAR scores candidate solutions by decomposing the joint log-likelihood of the reasoning trace and final answer into reasoning confidence and answer confidence components. The paper provides extensive empirical evaluation across several LLM/LRM families and challenging benchmarks, demonstrating that PiCSAR is both sample-efficient and outperforms established baselines  with fewer samples. Analyses highlight the utility and interpretability of the confidence signals, including novel perspectives via information planes and correlation between score dynamics and answer correctness.

**Strengths:**

1. PiCSAR appears to be straightforward to implement, requiring neither additional training nor an external reward model. Moreover, it can be seamlessly integrated into the reasoning workflow of any LLM or LRM, which highlights its practical utility and potential for broad adoption.

2. The manuscript is clearly written and presents well-structured arguments. The effectiveness of PiCSAR is convincingly demonstrated through extensive experimental validation. Furthermore, the use of an information plane visualization offers an interpretable perspective that enhances the reader’s understanding of the method’s underlying mechanisms.

**Weaknesses:**

1. The experimental evaluation primarily focuses on benchmarks where reasoning and answer extraction are relatively well-defined (e.g., mathematical reasoning, structured question answering). The generalization capability of PiCSAR to tasks involving open-ended question generation, instruction following, or scenarios with high answer uncertainty has not been examined.

2. The implementation of answer confidence relies on an “explicit instruction prompt” and specific answer extraction rules, which may introduce prompt dependency and parsing fragility.

**Questions:**

1. The manuscript employs joint log-likelihood as the re-ranking criterion. However, this approach may lead to the loss of a considerable amount of sequence information, which raises the question of whether such loss could influence the selection of the optimal reasoning chain. Could the authors elaborate on how PiCSAR mitigates this potential issue, and in particular, clarify why a straightforward summation of reasoning confidence and answer confidence can effectively distinguish the correct reasoning chain?

2. In addition to mathematical reasoning, has PiCSAR been evaluated on open-domain reasoning tasks? It would be valuable if the authors could further discuss the potential applicability of PiCSAR in other contexts, as well as any inherent limitations that may arise in such scenarios.

---

> ### Author Response · Authors · 2025-11-23
> **Responses to Reviewer MSGi (Part 1)**
>
> We thank reviewer MSGi for finding our framework easy to implement, and for appreciating the generalisability and extensive empirical evaluation towards LLMs and LRMs. We would like to offer clarifications and provide additional results that may address most of the concerns.
>
> > **W1** The experimental evaluation primarily focuses on benchmarks where reasoning and answer extraction are relatively well-defined [...] instruction following, or scenarios with high answer uncertainty has not been examined.
>
> Thank you for raising this concern! PiCSAR focuses on problems with well-defined reasoning boundaries,  and we believe this scope is both important and deliberate. Mathematical and scientific reasoning, where problems have well-defined reasoning and answer boundaries, represent a substantial class of real-world reasoning tasks. Furthermore, this setting enables a more accurate evaluation, currently infeasible for open-ended questions, which often admit multiple valid answers and make accuracy difficult to assess. This allows us to make precise claims about whether the model's confidence is a suitable metric for selecting the most accurate answer among generated samples.
>
> We acknowledge that extending to open-ended questions is an important next step. Incorporating a reward model to evaluate answers could be a promising direction to demonstrate broader applicability beyond fixed-format problems, and we look forward to exploring this in future work. Thank you again for raising this concern! We have clarified this point in the updated PDF, where we have added a `Limitations' section in Appendix D.
>
> > **W2** The implementation of answer confidence relies on an "explicit instruction prompt" [...] dependency and parsing fragility.
>
> Thank you for bringing this to our attention. We show that our results do not depend on the specific prompt and they are not subject to parsing fragility by running the *llama-3.1-8b* on the *MATH500 dataset* using 5 different prompts (listed below).
>
> Prompts:
> ```
> Prompt 1: *"You are a helpful assistant. When you see a potential partial reasoning followed by '<sep>', output the final answer. Here are some examples" + system_contents + "You are not allowed to provide any redundant symbols at for the final answer, including '#', '/'/, '$', '**' or others. Please only provide numbers as the final answer."*
>
> Prompt 2 (original prompt): "You are a helpful assistant. When you see a potential partial reasoning followed by '<sep>', output the final answer. Here are some examples"
>
> Prompt 3: “You are a helpful assistant. By providing the partial reasoning, output the final answer directly without any additional texts.”
>
> Prompt 4: “You are a helpful assistant. Based on the reasoning provided, output the final answer directly without any additional texts. Only Provide the final answer.”
>
> Prompt 5: “You are a helpful assistant. Provide the final answer directly without any additional texts (only the final answer) based on the partial reasoning.”
> ```
>
> | Prompt | Performance |
> |--------|-------------|
> | Prompt 1 | 54.60% |
> | Prompt 2 | 54.00% |
> | Prompt 3 | 54.20% |
> | Prompt 4 | 54.40% |
> | Prompt 5 | 54.40% |
>
> The accuracy remained consistent across all variations. This demonstrates that the specific prompt wording doesn't substantially affect the results. We have updated the results in the updated PDF version (Appendix B).

---

> > ### Author Response · Authors · 2025-11-23
> > **Responses to Reviewer MSGi (Part 2)**
> >
> > > **Q1**  The manuscript employs joint log-likelihood as the re-ranking criterion. However, this approach may lead to the loss of a considerable amount of sequence information [...] effectively distinguish the correct reasoning chain?
> >
> > We thank the reviewer for this question. Please note that our score corresponds to the joint probability $p(y,r \mid x)$ in log-space, which is equal to the sum of the two log-likelihoods: $\log p(r \mid x)+\log p(y \mid r,x)$. Since each term is computed token-wise, the score makes use of the entire reasoning sequence and the entire answer sequence, rather than discarding intermediate content. Thus, the summation simply reduces the two likelihood components to a single score, which can be used for ranking. We have included the clarification in line 192 of the updated PDF version.
> >
> > As shown in Figure 3, the median-defined quadrants already provide an effective decision heuristic, capturing 71.7% of correct predictions in the upper-right quadrant. Increasing the selectivity of the threshold (using the 75th percentile) yields an even clearer separation: for AIME, 100% of correct answers fall into this region (Appendix D.1), with further evidence provided in Appendix D.
> >
> >
> > By using joint probability, PiCSAR has a huge benefit relative to other methods, such as Self-Consistency (which only takes majority voting into account), Universal Self-Consistency, which identifies the majority consensus pattern rather than the correctness of the reasoning; these methods discard valuable signals from the reasoning process itself, which makes PiCSAR more reliable compared to the existing methods.
> >
> > > **Q2** In addition to mathematical reasoning, has PiCSAR […] that may arise in such scenarios.
> >
> > Please refer to our answer for **W1**.
> >
> > We hope our responses address your concerns. Please feel free to reach out with any further questions!

---

### Author Response · Authors · 2025-12-02
**Global Response**

We sincerely thank reviewers MSGi, dXEM, vZJv, and 8sjh, for their valuable feedback. We are really grateful for the constructive feedback and recognition of our framework!


**Main Contributions of the Paper**
- PiCSAR is a training-free probabilistic joint-likelihood framework to score and select reasoning chains that is straightforward and does not require any additional training nor an external reward model (MSGi, dXEM). PiCSAR is also generalisable towards LLMs and LRMs (MSGi)
- The thorough analysis of information plane visualization offers an interpretable perspective for PiCSAR, enhancing understanding of PiCSAR's underlying mechanisms.
- The proposed scoring function of PiCSAR provides a well-structured decomposition, potentially articulating the motivation for aggregating reasoning-related components in reasoning tasks.
- Last but not least, PiCSAR could be beneficial for researchers and practitioners, as it can easily be integrated into existing inference frameworks without additional compute for training or fine-tuning (dXEM).

**Experimental Results**
- PiCSAR is effective through extensive experimental validation, showing consistent performance gains across five LLMs and three LRMs against various baselines with systematic improvements (MSGi, dXEM, 8sjh, vZJv)
- PiCSAR's evaluations are impressive in that the improvements hold across different model sizes, reasoning and non-reasoning models, and a variety of benchmarks. (vZJv)

Last but not least, given that PiCSAR achieves performance improvement with almost no additional computation, it shows a great contribution in the context of growing interest in test-time scaling. (vZJv)

**Revised Manuscript**

Following the reviewers' feedback, We have revised our manuscript with multiple clarifications and experiments to strengthen the contributions of PiCSAR. All changes are highlighted in blue and extend to an additional page. The key additions are:

- Clarification on PiCSAR answer confidence. We have clarified the answer confidence fallback mechanism using $log p(y \mid r,X) =-10$ accompanied by ablation studies across multiple fallback threshold values.
- Statistical analysis of PiCSAR's empirical performance. We have added Nemenyi tests on the experimental results, demonstrating - PiCSAR's statistical significance and consistency against multiple baselines.
- Clarifications on experimental setup. We provide detailed clarifications on experimental configurations to enhance reproducibility and transparency.
- Details and distinction from existing baselines. We have expanded the explanation of all baselines and clarified PiCSAR's conceptual and empirical novelty, strengthening our contribution statement.
- Ablation studies on prompts for answer confidence. We include ablation studies across five different prompts for extracting answer confidence, demonstrating consistent accuracy.
- Analysis of token length, reasoning confidence, and accuracy. We provide a comprehensive analysis examining the relationships between these three factors.
- Clarification between length-normalized and non-normalized variants. We clarify the empirical similarities between PiCSAR (non-length normalised) and PiCSAR-N (length-normalised).
- Ablation studies on sample size and temperature. We present additional ablation studies evaluating performance across different numbers of samples and temperature settings.
- Qualitative examples. We provide additional qualitative examples illustrating PiCSAR's scoring mechanism.
Limitations section. We have added a dedicated section discussing PiCSAR's limitations.

We have addressed all individual reviewer concerns in our point-by-point responses, addressing their specific concerns and questions. We sincerely thank the reviewers for all the feedback that has made the revised manuscript substantially stronger.

---

### Note · Authors · 2026-01-03

I have read and agree with the venue's withdrawal policy on behalf of myself and my co-authors.